JCB Journal of Cell Biology

## TOOLS

# The TMEM192-mKeima probe specifically assays lysophagy and reveals its initial steps

Takayuki Shima[1], Monami Ogura[2], Ruriko Matsuda[2], Shuhei Nakamura[1,2,3], Natsuko Jin[1], Tamotsu Yoshimori[1,2,4], and Akiko Kuma[1]

**Membrane rupture of lysosomes results in leakage of their contents, which is harmful to cells. Recent studies have reported that several systems contribute to the repair or elimination of damaged lysosomes. Lysophagy is a type of selective autophagy that plays a crucial role in the lysosomal damage response. Because multiple pathways are involved in this response, an assay that specifically evaluates lysophagy is needed. Here, we developed the TMEM192-mKeima probe to evaluate lysophagy. By comparing the use of this probe with the conventional galectin-3 assay, we showed that this probe is more specific to lysophagy. Using TMEM192-mKeima, we showed that TFEB and p62 are important for the lysosomal damage response but not for lysophagy, although they have previously been considered to be involved in lysophagy. We further investigated the initial steps in lysophagy and identified UBE2L3, UBE2N, TRIM10, 16, and 27 as factors involved in it. Our results demonstrate that the TMEM192-mKeima probe is a useful tool for investigating lysophagy.**

## Introduction

Lysosomes are intracellular organelles responsible for the degradation of various cellular components, including proteins, nucleic acids, and lipids (Saftig and Klumperman, 2009). These macromolecules enter lysosomes by various routes, such as endocytosis or autophagic pathways (Luzio et al., 2009; Mizushima et al., 2008), and the degradation products can subsequently be reused for cellular metabolic processes. Lysosomes are characterized by an acidic internal pH, which allows numerous acid hydrolases to degrade their substrates (Ballabio and Bonifacino, 2020). Lysosomes are involved in various cellular functions, and the collapse of lysosomal homeostasis underlies the pathogenesis of a number of diseases (Bonam et al., 2019). The lysosomal membrane rupture is induced by various intracellular and extracellular factors, such as oxidative stress, lipids, silica, monosodium urate, bacterial toxins, and β-amyloid (Papadopoulos and Meyer, 2017; Cantuti-Castelvetri et al., 2018; Mossman and Churg, 1998; Gómez-Sintes et al., 2016). This membrane rupture results in leakage of lysosomal contents, which is deleterious for cells and can lead to cell death in extreme cases (Wang et al., 2018).

Recent studies have reported that several systems contribute to the repair or elimination of damaged lysosomes, including selective autophagy (lysophagy), membrane repair mediated by endosomal sorting complexes required for transport (ESCRT),

and transcriptional regulation by transcription factor EB (TFEB; Papadopoulos et al., 2020). In addition, very recent studies have demonstrated that damaged lysosomes are repaired by annexins, a group of calcium-responding membrane repair proteins, sphingomyelin exposure on the cytosolic surface of damaged lysosomes, and lipid transport from the ER via ER–lysosome membrane contact sites (Tan and Finkel, 2022; Radulovic et al., 2022; Yim et al., 2022; Niekamp et al., 2022).

In autophagy, which is a conserved intracellular degradation pathway in eukaryotes, a double-membrane vesicle called the autophagosome sequesters cytoplasmic components such as proteins and organelles, which are then degraded after the autophagosome fuses with a lysosome (Bento et al., 2016). Lysophagy removes severely damaged lysosomes, which contributes to maintaining lysosomal homeostasis (Hung et al., 2013; Maejima et al., 2013). The mechanisms in the initial steps of lysophagy, especially ubiquitination of damaged lysosomes, which is an important signal to drive lysophagy, are still being elucidated. In addition to lysophagy, ESCRT components are involved in the response to lysosomal damage. Recent studies have shown that ALIX and TSG101 are recruited to damaged lysosomes, where they repair lysosomal membranes with relatively small ruptures (Skowyra et al., 2018; Radulovic et al., 2018). Transcriptional regulation by TFEB is also required for the response to lysosomal

[1]Department of Genetics, Graduate School of Medicine, Osaka University, Osaka, Japan; [2]Department of Intracellular Membrane Dynamics, Graduate School of Frontier Biosciences, Osaka University, Osaka, Japan; [3]Institute for Advanced Co-Creation Studies, Osaka University, Osaka, Japan; [4]Integrated Frontier Research for Medical Science Division, Institute for Open and Transdisciplinary Research Initiatives (OTRI), Osaka University, Osaka, Japan.

Correspondence to Akiko Kuma: kuma@fbs.osaka-u.ac.jp; Tamotsu Yoshimori: tamyoshi@fbs.osaka-u.ac.jp;

N. Jin's current affiliation is the Live Cell Super-Resolution Imaging Research Team, RIKEN Center for Advanced Photonics, Wako, Japan.

damage and contributes to maintaining lysosomal homeostasis (Nakamura et al., 2020; Chauhan et al., 2016; Jia et al., 2018). TFEB, a master regulator of lysosomal biogenesis and autophagy at the transcriptional level (Settembre et al., 2011; Sardiello et al., 2009), is activated by calcium efflux from lysosomes that occurs in response to lysosomal damage; this activation is mediated by non-canonical LC3 lipidation of damaged lysosomes (Nakamura et al., 2020).

Galectin 3 (Gal-3), a β-galactose-binding lectin, has been used as a marker protein to monitor lysosomal membrane rupture (Paz et al., 2010). We previously developed a Gal-3 clearance assay to evaluate lysophagic activity (Maejima et al., 2013). Gal-3 normally localizes to the cytoplasm, and lysosomal membrane damage enables Gal-3 to access galactoses in lysosomes. Consequently, damaged lysosomes labeled with Gal-3 are observed as puncta under fluorescence microscopy, and a reduction in the number of these puncta reflects recovery from damage as a result of lysophagy (Maejima et al., 2013). At present, the Gal-3 clearance assay is the approach most commonly used to measure lysophagy activity. However, this method evaluates recovery from damage rather than directly assessing lysophagy. Thus, there is concern that the assay may reflect not only lysophagy but also other lysosomal damage responses such as ESRCT-mediated repair and TEEB-mediated transcription. This prompted us to establish an assay specific for lysophagy. In this study, we developed a lysophagy flux assay using the TMEM192-mKeima probe, which allowed us to assess lysophagic activity separately from other pathways involved in the lysosomal damage response. Taking advantage of this, we showed that TFEB and p62, which have been considered to be involved in lysophagy, are important for the lysosomal damage response but not for lysophagy. In addition, we identified UBE2L3, UBE2N, and TRIM10, 16, and 27 as lysophagy regulators. Altogether, we successfully developed the TMEM192-mKeima probe as a useful tool for investigating lysophagy.

## Results

### Development of the TMEM192-mKeima probe for monitoring lysophagy

The mKeima fluorescent protein has been widely used as a probe for monitoring the delivery of autophagy substrates into lysosomes (Katayama et al., 2011). mKeima is excited by light at a wavelength of 440 nm at neutral pH; when it is exposed to an acidic environment, the excitation wavelength changes to 590 nm. Thus, the ratio of mKeima excitation wavelengths at 590 and 440 nm reflects the delivery of mKeima from the cytoplasm (neutral) into lysosomes (acidic), thereby measuring autophagic activity. To apply mKeima to lysophagy, we fused mKeima at the C-terminus of the lysosomal membrane protein TMEM192. TMEM192 is a commonly used lysosomal marker and has been used for imaging and immunoprecipitation of lysosomes (Abu-Remaileh et al., 2017). As shown in the scheme in Fig. 1 A, TMEM192-mKeima localizes to the lysosomal membrane and is exposed to the cytoplasm under normal conditions. After induction of lysosomal membrane damage by the lysosomotropic agent L-leucyl-L-leucine methyl ester (LLOMe), damaged lysosomes

labeled with TMEM192-mKeima are sequestered by autophagosomes, which then fuse with intact lysosomes, resulting in the accumulation of TMEM192-mKeima in autolysosomes/lysosomes, which are acidic compartments. Consequently, lysophagic activity can be measured by comparing the fluorescence intensity of TMEM192-mKeima at excitation wavelengths of 590 and 440 nm, which indicates their presence inside and outside of lysosomes, respectively (Fig. 1 A).

To demonstrate the feasibility of this system, we first confirmed the localization of TMEM192-mKeima in HeLa cells that stably expressed TMEM192-mKeima. TMEM192-mKeima exhibited punctate structures and colocalized with lysosomal-associated membrane protein 1 (LAMP1), confirming that TMEM192-mKeima properly targeted lysosomes (Fig. S1 A). We next measured the fluorescence intensity of TMEM192-mKeima at excitation wavelengths of 445 and 594 nm by live cell imaging. After 1 h of treatment with LLOMe, cells were cultured for an additional 10 h in the absence of LLOMe. TMEM192-mKeima signals were observed over the time course (Fig. 1 B and Fig. S2 A). Weak TMEM192-mKeima signals at 594 nm excitation were already observed before LLOMe treatment (Fig. 1 B and Fig. S2 A). Those signals may represent the basal level of lysophagy or invagination of the lysosomal membrane through some other machinery such as multivesicular body (MVB) formation/microautophagy. The basal level puncta disappeared with LLOMe treatment, probably because of the loss of acidity of lysosomes due to the drug (Fig. S2 A). During recovery from lysosomal damage, TMEM192-mKeima puncta at 594 nm excitation was more evident than at basal levels and gradually increased in number with incubation time (Fig. 1 B and Fig. S2 A). We measured the number of TMEM192-mKeima puncta in acidic compartments during lysosomal damage (Fig. 1 C and Fig. S2 A). For this quantification, we carefully set a threshold for each experiment because TMEM192-mKeima in lysosomes showed weak signals at 445 nm excitation, and intact lysosomes that contain other damaged lysosomes also have their own TMEM192-mKeima signals at 445 nm excitation. The intensity at 594 nm was compared with the intensity at 445 nm, and if this ratio was above the set threshold, the puncta of mKeima were considered to be TMEM192-mKeima in an acidic environment. The number of TMEM192-mKeima puncta in lysosomes was increased with incubation time after LLOMe treatment and also positively correlated with the concentration of LLOMe (Fig. 1 C and Fig. S2 B). Occasionally, TMEM192-mKeima signals at 594 nm excitation were observed inside the ring-like structure of TMEM192-mKeima signals at 445 nm excitation, indicating the presence of lysosomal membrane within another lysosome (Fig. 1 D). When lysosomes were neutralized by treatment with bafilomycin A1, the TMEM192-mKeima puncta in lysosomes was abolished (Fig. S3 A). These results suggest that the TMEM192-mKeima system successfully monitored the delivery of lysosomes into other lysosomes in response to LLOMe treatment.

To examine whether the appearance of TMEM192-mKeima puncta in lysosomes following LLOMe treatment depended on autophagy, we measured the 594/445 nm ratio in autophagy-deficient cells. We tested knockdown of *FIP200*, *ATG14*, *ATG7*, *ATG12*, *ATG16*, or syntaxin (*STX*)17. ATG proteins play roles in

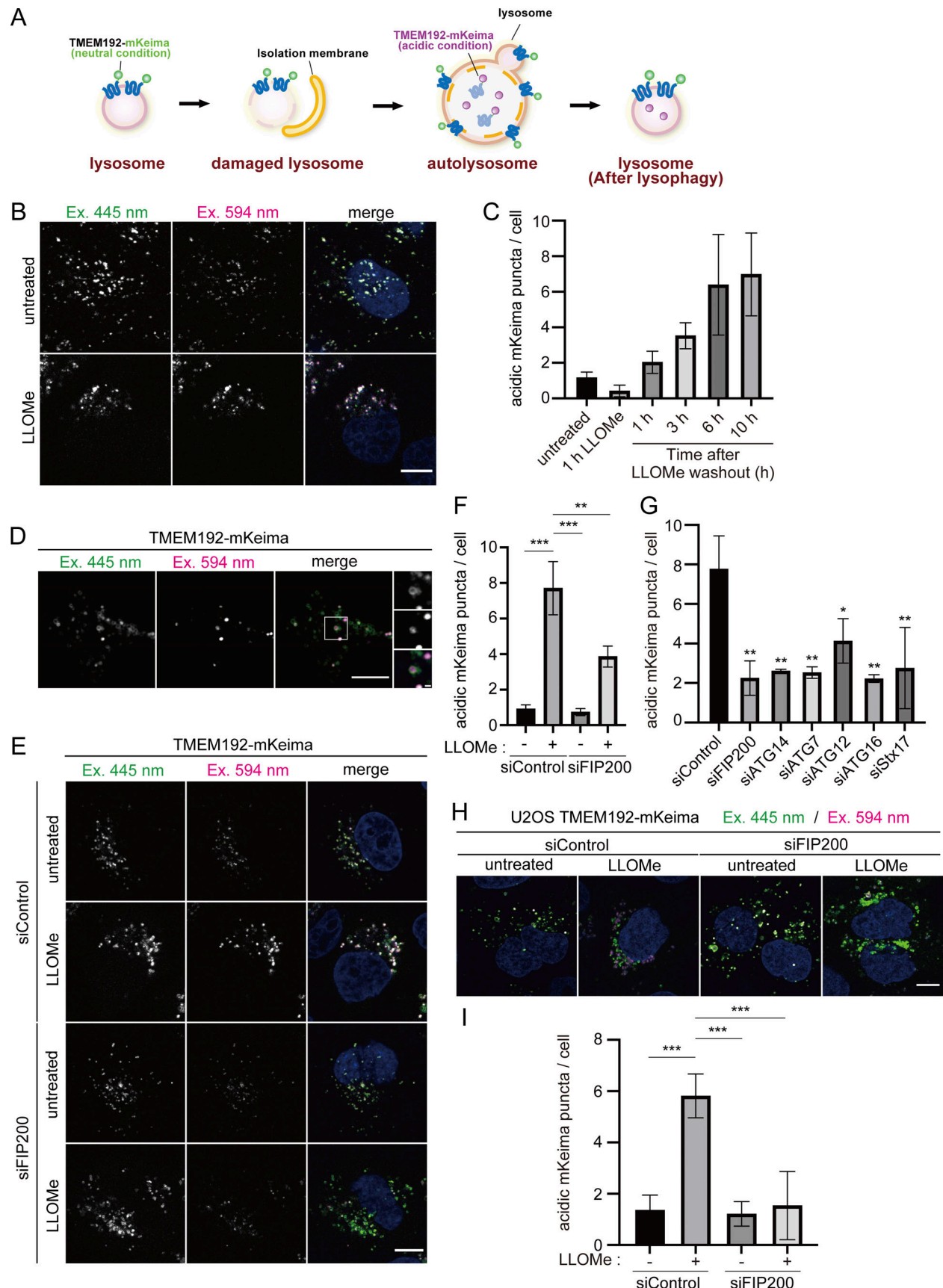

Figure 1. **Establishment of a lysophagy flux assay using TMEM192-mKeima. (A)** Schematic diagram of the lysophagy assay using TMEM192-mKeima. **(B)** HeLa cells stably expressing TMEM192-mKeima were cultured for 1 h in the presence of 1 mM LLOMe, followed by culture in DMEM containing Hoechst

without LLOMe. After incubation for 10 h, cells were observed under a fluorescence microscope. Scale bars, 10 µm. **(C)** Quantification of the number of acidic TMEM192-Keima puncta per cell for Fig. S2 A. >30 cells were analyzed per condition in each experiment. Error bars indicate means ± SEM in at least three independent experiments (*n* = 3). **(D)** HeLa cells stably expressing TMEM192-mKeima were cultured for 1 h in the presence of 1 mM LLOMe, followed by culture in DMEM containing Hoechst without LLOMe. Cells were observed using a 100× oil immersion objective under a fluorescence microscope. Scale bars, 10 µm. **(E)** HeLa cells stably expressing TMEM192-mKeima were treated with siRNA against FIP200 or treated with control siRNA. After 48 h of siRNA treatment, cells were cultured for 1 h in the presence of 1 mM LLOMe. After LLOMe was removed, cells were cultured in DMEM containing Hoechst for 10 h and observed by fluorescence microscopy. Scale bars, 10 µm. **(F)** Quantification of the number of acidic mKeima puncta per cell in cells for Fig. 1 E. >30 cells were analyzed per condition in each experiment. Error bars indicate means ± SEM in at least three independent experiments (*n* = 3). ***P < 0.001 (one-way ANOVA followed by Dunnett's test). **(G)** HeLa cells stably expressing TMEM192-mKeima were treated with siRNA against FIP200, ATG14, ATG7, ATG12, ATG16, or Stx17, or treated with control siRNA. After 72 h of siRNA treatment, cells were cultured for 1 h in the presence of 1 mM LLOMe. After LLOMe was removed, cells were cultured in DMEM containing Hoechst for 10 h and observed by fluorescence microscopy. Scale bars, 10 µm. The graph shows the number of acidic TMEM192-mKeima puncta per cell in these HeLa cells. >30 cells were analyzed per condition in each experiment. Error bars indicate means ± SEM in at least three independent experiments (*n* = 3). *P < 0.05; **P < 0.01 (one-way ANOVA followed by Dunnett's test). **(H)** U2OS cells stably expressing TMEM192-mKeima were treated with siRNA against FIP200 or treated with control siRNA. After 48 h of siRNA treatment, cells were cultured for 1 h in the presence of 1 mM LLOMe. After LLOMe was removed, cells were cultured in DMEM containing Hoechst for 6 h and observed by fluorescence microscopy. Scale bars, 10 µm. **(I)** Quantification of the number of acidic mKeima puncta per cell in cells for Fig. 1 H. More than 30 cells were analyzed per condition in each experiment. Error bars indicate means ± SEM in at least three independent experiments (*n* = 3). ***P < 0.001 (one-way ANOVA followed by Dunnett's test).

autophagosome formation, and STX17 is a SNARE protein that is required for autophagosome formation and autophagosome–lysosome fusion (Itakura et al., 2012; Hamasaki et al., 2013). In control cells, the number of mKeima puncta in lysosomes was increased by LLOMe treatment, whereas there were significantly fewer of them in FIP200-knockdown cells (Fig. 1, E and F). The presence of TMEM192-mKeima puncta in lysosomes was also suppressed in the cells with knockdown of other ATGs and STX17, suggesting that the TMEM192-mKeima signals in lysosomes induced by LLOMe represent lysophagic activity (Fig. 1 G). Similar results were obtained from the experiments using *ATG*-knockout cells (Fig. S3 B). To determine whether TMEM192-mKeima assay is applicable to other cell lines besides HeLa cells, we tested U2OS cells, which are also commonly used in autophagy research. TMEM192-mKeima puncta in lysosomes was increased when lysosomes were damaged, and this increase was suppressed by the knockdown of FIP200 (Fig. 1, H and I). Taking these findings together, we succeeded in developing a lysophagy flux assay using the TMEM192-mKeima probe.

## TMEM192-mKeima probe can assess lysophagic activity separately from other lysosomal damage responses

Gal-3 has been widely used as a marker for monitoring lysosomal membrane rupture (Paz et al., 2010; Maejima et al., 2013). In the Gal-3 clearance assay, a reduction in the number of damaged lysosomes (Gal-3-positive lysosomes) is considered to indicate recovery from lysosomal damage through lysophagy. However, this assay may reflect not only lysophagy but also other lysosomal damage responses, including repair by ESCRT and transcriptional regulation by TFEB. To address this issue, we compared the Gal-3 clearance assay with the use of the TMEM192-mKeima system in cells expressing both probes. We first observed a correlation between a decrease in GFP-Gal3 puncta and an increase in lysophagic puncta labeled with TMEM192-mKeima under conditions of lysosomal damage (Fig. 2, A and B). Cells stably expressing both GFP-Gal3 and TMEM192-mKeima were treated with LLOMe for 1 h. After the reagent was washed out, the cells were cultured for an additional 10 h in the absence of LLOMe. The numbers of GFP-Gal3 and TMEM192-mKeima puncta in lysosomes were counted at

the indicated timepoints. GFP-Gal3 puncta appeared during the 1 h LLOMe treatment and continuously and gradually decreased until the end of the assay at 10 h after washout (Fig. 2, A and B). The loss of GFP-Gal3 puncta was clearly delayed in FIP200-knockdown cells, confirming that the reduction of GFP-Gal3 puncta was dependent on autophagy (Fig. 2, A and B). However, some reduction of GFP-Gal3 dots was still observed in FIP200-knockdown cells, indicating that the Gal-3 clearance assay may monitor other lysosomal damage responses in addition to lysophagy. Inversely correlated with the reduction of GFP-Gal3 puncta, lysophagic TMEM192-mKeima puncta were evident at 3 h after washout and gradually increased in number with incubation time, which was strongly suppressed in FIP200-knockdown cells (Fig. 2, A and B).

To further examine the advantages of the TMEM192-mKeima probe, we next compared Gal-3 assay with TMEM192-mKeima assay in cells lacking ESCRT or TFEB. ESCRT and TFEB were previously reported to be involved in the lysosomal damage response (Skowyra et al., 2018; Radulovic et al., 2018; Nakamura et al., 2020). Although the ESCRT machinery has been reported to participate in autophagosome closure (Takahashi et al., 2018), suppression of Alix, a component known to play a role in the early steps of activity of the ESCRT machinery, was found to have no effect on autophagy, but it did delay the recovery of damaged lysosomes (Murrow et al., 2015; Skowyra et al., 2018; Petiot et al., 2008). We therefore chose to knock down Alix to suppress lysosomal membrane repair by ESCRT without affecting lysophagy. While Alix-knockdown cells expressing both probes showed a delay in the clearance of Gal-3 puncta (Fig. 2, C and D), there was no effect on the presence of lysophagic TMEM192-mKeima puncta (Fig. 2 E and Fig. S4 A). This is consistent with previous reports showing that membrane repair mediated by the ESCRT machinery occurs independently of lysophagy (Skowyra et al., 2018; Radulovic et al., 2018). These results suggest that the Gal-3 clearance assay reflects not only lysophagy but also membrane repair mediated by ESCRT.

Taking advantage of the specificity of TMEM192-mKeima assay to lysophagy, we examined the effect of TFEB knockdown on lysophagy. Since TFEB is a master regulator of autophagy gene expression, it has been indicated that TFEB is

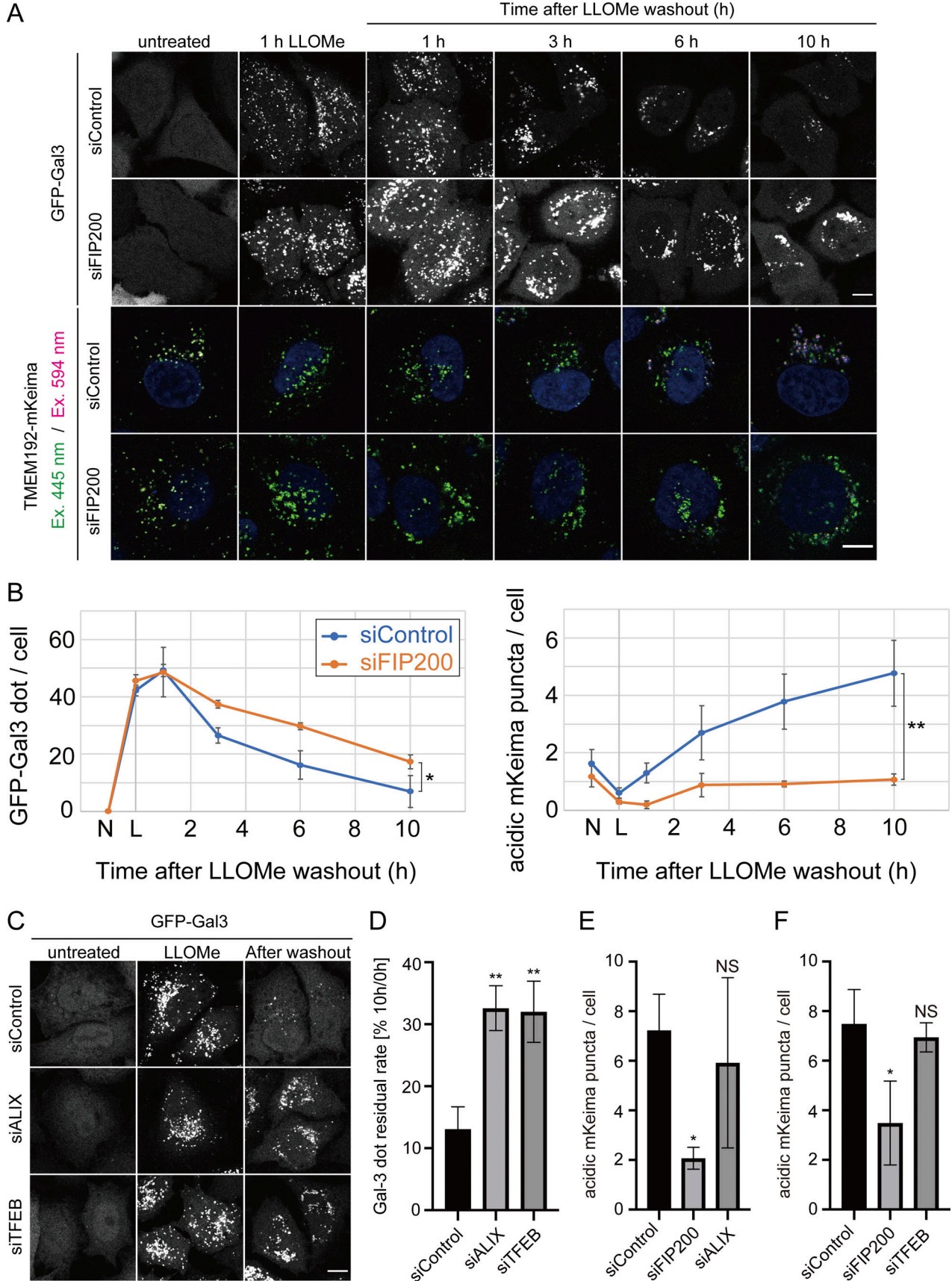

Figure 2. **TMEM192-mKeima can assess lysophagic activity separately from ESCRT-mediated and TFEB-mediated lysosomal damage recovery.** **(A)** HeLa cells stably expressing both TMEM192-mKeima and GFP-Gal3 were treated with siRNA against FIP200 or treated with control siRNA. After 48 h of

siRNA treatment, cells were cultured for 1 h in the presence of 1 mM LLOMe, followed by culture in DMEM containing Hoechst without LLOMe. After incubation for 1, 3, 6, or 10 h, cells were observed under a fluorescence microscope. Scale bars, 10 µm. **(B)** The left graph shows the number of Gal-3 puncta per cell for Fig. 2 A. The right graph shows the number of acidic mKeima puncta per cell for Fig. 2 A. N indicates no drug treatment and L indicates 1 h of LLOMe treatment. More than 30 cells were analyzed per condition in each experiment. Error bars indicate means ± SEM in at least three independent experiments (n = 3). *P < 0.05; **P < 0.01 (unpaired two-tailed Student's t test). **(C)** HeLa cells stably expressing GFP-Gal3 were treated with siRNA against Alix, TFEB, or treated with control siRNA. After 48 h of siRNA treatment, cells were cultured for 1 h in the presence of 1 mM LLOMe, followed by culture in DMEM without LLOMe. After 10 h, cells were observed by fluorescence microscopy. Scale bars, 10 µm. **(D)** Quantification of the percentage of Gal-3 puncta remaining after 10 h of incubation for Fig. 2 C. More than 80 cells were analyzed per condition in each experiment. Error bars indicate means ± SEM in at least three independent experiments (n = 3). **P < 0.01 (one-way ANOVA followed by Dunnett's test). **(E and F)** Quantification of the number of acidic mKeima puncta per cell in cells for Fig. S4, A and B. More than 30 cells were analyzed per condition in each experiment. Error bars indicate means ± SEM in at least three independent experiments (n = 3). *P < 0.05, NS: not significant (one-way ANOVA followed by Dunnett's test).

involved in the lysosomal damage response by regulating lysophagy. Clearance of Gal-3 puncta was delayed by TFEB suppression, as previously reported (Skowyra et al., 2018; Radulovic et al., 2018; Fig. 2, C and D), while the number of lysophagic TMEM192-mKeima puncta was not affected by TFEB knockdown (Fig. 2 F and Fig. S4 B). These findings suggest that TFEB is important for the lysosomal damage response but not through lysophagy. To further confirm that the GFP-Gal3 assay monitors multiple lysosomal damage response pathways, we compared the effect of knockdown of FIP200 (lysophagy), Alix (ESCRT pathway), TFEB (TFEB pathway), or all three of them (Fig. S4 C). HeLa cells stably expressing GFP-Gal3 treated with each siRNA were cultured with 1 mM LLOMe for 1 h, followed by culture without LLOMe for 10 h. The clearance of Gal-3 dots was suppressed in cells knocking down FIP200, Alix, or TFEB to the same degree (Fig. S4 C). Their triple knockdown was strongly suppressed the clearance of damaged lysosomes, but not completely, indicating other pathways also may be monitored (Fig. S4 C). Taken together, these findings indicate that the TMEM192-mKeima system successfully distinguishes lysophagy from other lysosomal damage responses compared with the Gal-3 assay.

### Clarifying autophagy adaptors required for lysophagy
Using the TMEM192-mKeima system, we explored the initial steps of lysophagy to elucidate the underlying mechanism. First, we focused on autophagy adaptors. In selective autophagy for specific cargo, including the endoplasmic reticulum (ER), mitochondria, peroxisomes, pathogens, and protein aggregates, cargo surface proteins are often ubiquitinated to recruit autophagy adaptor proteins such as p62, OPTN, NDP52, TAX1BP1, and NBR1 (Johansen and Lamark, 2020). These autophagy adaptors are currently thought to be required both for cargo recognition via association with ubiquitin and for subsequent sequestration by autophagosomes via binding to LC3 family proteins on the autophagosomal membrane. The importance of each adaptor depends on the cargo (Johansen and Lamark, 2020). For example, in selective autophagy for mitochondria, OPTN and NDP52 are critical for Parkin-mediated mitophagy (Lazarou et al., 2015).

Previous studies using Gal-3 and LC3 as probes showed that p62, OPTN, and TAX1BP1 are required for lysophagy (Papadopoulos et al., 2017; Eapen et al., 2021; Gallagher and Holzbaur, 2023); however, conflicting findings have been reported regarding the involvement of p62. Because the assays

using Gal-3 and LC3 as probes may reflect the responses to lysosomal damage other than lysophagy as described above and reported previously (Nakamura et al., 2020), we re-evaluated the involvement of autophagy adaptors using the TMEM192-Keima system. We first observed the localization of autophagy adaptors, namely, p62, OPTN, NDP52, TAX1BP1, and NBR1, during lysophagy. We observed endogenous p62 and other exogenous adaptors tagged with green fluorescent protein (GFP). As in previous studies, the puncta of endogenous p62 was highly apparent after LLOMe treatment and localized to lysosomes (Fig. S5 A). The other four adaptors tagged with GFP were also observed as puncta upon LLOMe treatment and most of them co-localized with LAMP1 (Fig. S5 B). These findings suggest that all adaptors have the ability to localize to damaged lysosomes.

Next, we investigated which autophagy adaptors are required for lysophagy. We measured lysophagic activity using TMEM192-mKeima in HeLa cells lacking the five autophagy adaptors (hereafter referred to as Penta-KO cells; Lazarou et al., 2015). In contrast to WT cells, the presence of lysophagic TMEM192-mKeima puncta in LLOMe-treated cells was markedly suppressed in Penta-KO cells, suggesting that autophagy adaptors are required for lysophagy (Fig. 3 A). Consistent with this result, there was a clear delay in the recovery of lysosomal activity in Penta-KO cells judging from the activity of lysosomal enzyme cathepsin B (Fig. S6). Because of the conflicting findings regarding the involvement of p62 in lysophagy (Papadopoulos et al., 2017; Eapen et al., 2021; Gallagher and Holzbaur, 2023), we investigated whether p62 is required for lysophagy by comparing TMEM192-mKeima assay with Gal-3 assay. Knockdown of p62 did not affect lysophagy as determined by the TMEM192-mKeima assay (Fig. 3 B), but resulted in defective clearance of Gal-3 puncta (Fig. 3 C). These results suggest that p62 may function in lysosomal damage responses other than lysophagy.

We then reintroduced each adaptor into Penta-KO (knock out five autophagy adaptors) cells expressing TMEM192-mKeima and measured LLOMe-stimulated lysophagic flux. Consistent with the previous results, TMEM192-mKeima puncta in lysosomes were not detected in Penta-KO cells re-expressing FLAG-p62 or FLAG-NBR1, whereas the puncta were observed after the re-expression of FLAG-OPTN, FLAG-NDP52, and FLAG-TAX1BP1 under conditions with lysosomal damage (Fig. 3, D and E). These results suggest that OPTN, NDP52, and TAX1BP1 play crucial roles in lysophagy. Conversely, knockdown of each of these three adaptors resulted in partial defects in lysophagy, and suppression of all three adaptors resulted in almost complete

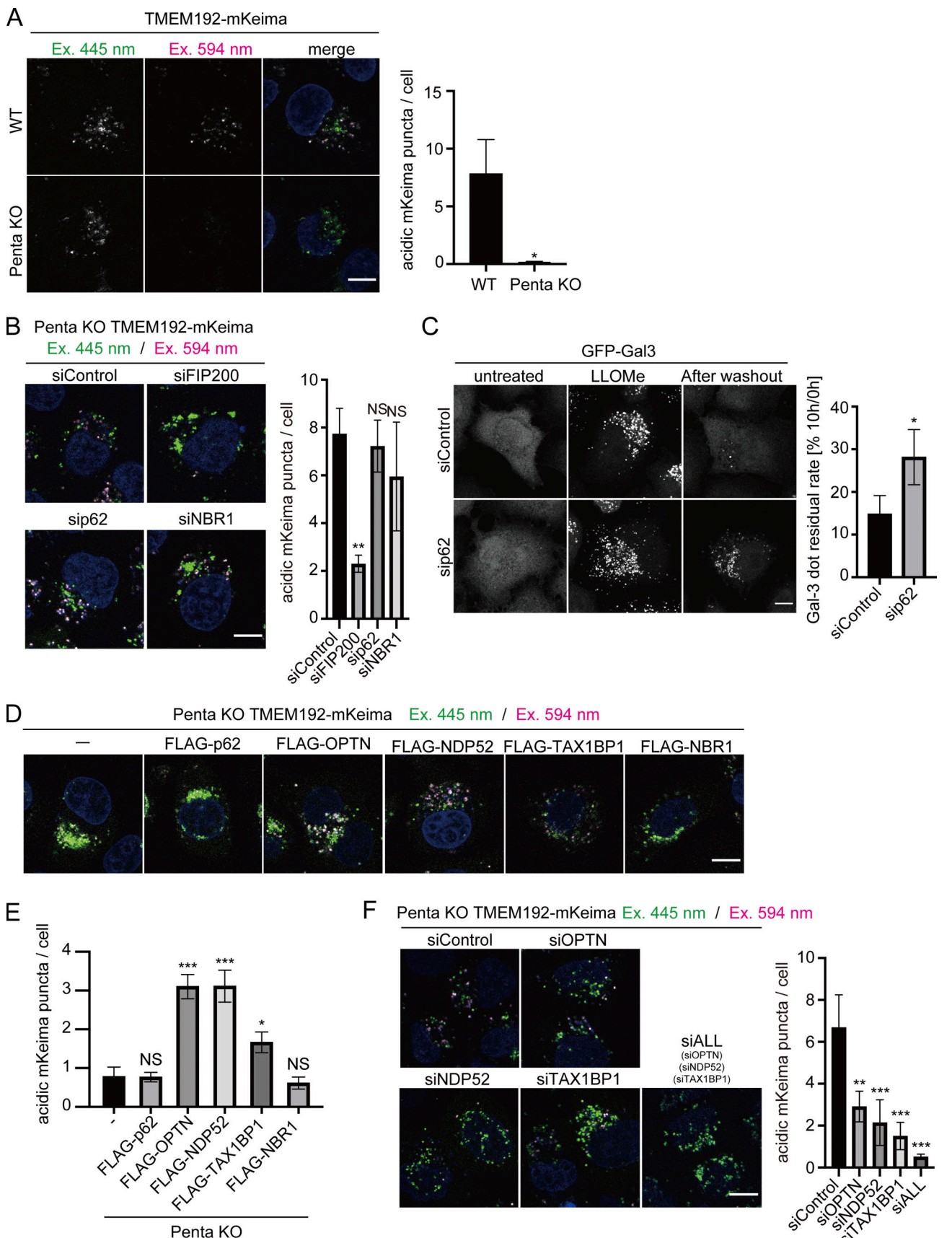

Figure 3. **OPTN, NDP52, and TAX1BP1 play crucial roles in lysophagy. (A)** WT and Penta-KO (p62, OPTN, NDP52, TAX1BP1, and NBR1 KO) HeLa cells stably expressing TMEM192-mKeima were cultured for 1 h in the presence of 1 mM LLOMe. After LLOMe was removed, cells were cultured in DMEM containing

Hoechst for 10 h and then analyzed by fluorescence microscopy. Scale bars, 10 μm. The graph shows the number of acidic TMEM192-mKeima puncta per cell in WT and Penta-KO HeLa cells. More than 30 cells were analyzed per condition in each experiment. Error bars indicate means ± SEM in at least three independent experiments ($n$ = 3). **P < 0.01 (unpaired two-tailed Student's $t$ test). **(B)** HeLa cells stably expressing TMEM192-mKeima were treated with two siRNAs against each of p62, NBR1, FIP200, or control siRNA. After 48 h of siRNA treatment, cells were cultured for 6 h in the presence of 1 mM LLOMe in DMEM containing Hoechst and observed by fluorescence microscopy. Scale bars, 10 μm. The graph shows the number of acidic TMEM192-mKeima puncta per cell in each cell. >30 cells were analyzed per condition in each experiment. Error bars indicate means ± SEM in at least three independent experiments ($n$ = 3). **P < 0.01, NS: not significant (one-way ANOVA followed by Dunnett's test). **(C)** HeLa cells stably expressing GFP-Gal3 were treated with siRNA against p62 or treated with control siRNA. After 48 h of siRNA treatment, cells were cultured for 1 h in the presence of 1 mM LLOMe, followed by culture in DMEM without LLOMe. After 10 h, cells were observed by fluorescence microscopy. Scale bars, 10 μm. Quantification of the percentage of Gal-3 puncta remaining after 10 h of incubation. >80 cells were analyzed per condition in each experiment. Error bars indicate means ± SEM in at least three independent experiments ($n$ = 3). *P < 0.05 (unpaired two-tailed Student's $t$ test). **(D)** Penta-KO cells stably expressing TMEM192-mKeima and each FLAG-adaptor (p62, OPTN, NDP52, TAX1BP1, and NBR1) were cultured for 1 h in the presence of 1 mM LLOMe. After LLOMe was removed, cells were cultured in DMEM containing Hoechst for 10 h and then analyzed by fluorescence microscopy. Scale bars, 10 μm. **(E)** Quantification of the number of acidic TMEM192-mKeima puncta per cell for Fig. 3 D. >30 cells were analyzed per condition in each experiment. Error bars indicate means ± SEM in at least three independent experiments ($n$ = 3). *P < 0.05; ***P < 0.001, NS: not significant (one-way ANOVA followed by Dunnett's test). **(F)** HeLa cells stably expressing TMEM192-mKeima were treated with two siRNAs against each of OPTN, NDP52, TAX1BP1, or control siRNA. After 48 h of siRNA treatment, cells were cultured for 6 h in the presence of 1 mM LLOMe in DMEM containing Hoechst and observed by fluorescence microscopy. Scale bars, 10 μm. The graph shows the number of acidic TMEM192-mKeima puncta per cell in each cell. >30 cells were analyzed per condition in each experiment. Error bars indicate means ± SEM in at least three independent experiments ($n$ = 3). ***P < 0.001 (one-way ANOVA followed by Dunnett's test).

---

loss of lysophagy (Fig. 3 F). Taking these findings together, although all five autophagy adaptors were recruited to damaged lysosomes during lysophagy, our data suggested that OPTN, NDP52, and TAX1BP1 are important for lysophagy.

### TANK-binding kinase 1 (TBK1) is recruited to damaged lysosomes in a calcium-dependent manner and is required for lysophagy

Next, we explored the upstream of autophagy receptors. A previous study showed that TBK1 is required for lysophagy (Eapen et al., 2021). TBK1 is an inducer of type 1 interferons and is classified as a member of the IKK kinase family that is involved in innate immune signaling pathways (Fitzgerald et al., 2003; Ahmad et al., 2016). TBK1 also plays a role in bulk autophagy and adaptor-mediated selective autophagy of pathogens and depolarized mitochondria (Pilli et al., 2012; Thurston et al., 2009; Kumar et al., 2019). Previous studies showed that TBK1 directly phosphorylates p62, OPTN, NDP52, and TAX1BP1 and thus promotes the recognition and sequestration of autophagosome substrates by facilitating their association with ubiquitin and LC3, respectively (Richter et al., 2016; Pilli et al., 2012; Minowa-Nozawa et al., 2017; Heo et al., 2015). TBK1 is activated by self-oligomerization and trans-autophosphorylation at residue Ser172 (Larabi et al., 2013; Tu et al., 2013; Shu et al., 2013). The TMEM192-mKeima system showed that lysophagic flux was markedly suppressed in TBK1-knockdown cells compared with that in control cells (Fig. 4, A and B), which was consistent with a previous study using Gal-3 as a marker (Eapen et al., 2021). We also confirmed that GFP-TBK1 and p-TBK1 were present mostly throughout the cytoplasm, whereas some puncta appeared on lysosomes after LLOMe treatment (Fig. 4, C and D). To understand further upstream of the TBK1 signaling, we investigated whether the targeting of TBK1 to lysosomes depends on Ca²⁺. This approach was taken because several proteins involved in the responses to lysosomal damage, such as the ESCRT components, LC3s, and annexins, have been shown to target damaged lysosomes in a manner dependent on calcium leaking from the lysosomes (Skowyra et al., 2018; Nakamura et al., 2020; Yim et al., 2022). HeLa cells were treated with both LLOMe and

BAPTA, an intracellular Ca²⁺ chelator. Recruitment of pTBK1 to lysosomes in response to LLOMe was abolished by BAPTA treatment (Fig. 4, D and E). Meanwhile, Western blotting showed that TBK1 was phosphorylated by LLOMe treatment, and this phosphorylation was not inhibited by BAPTA treatment, suggesting that the targeting of TBK1 to lysosomes but not its activation was dependent on calcium signals (Fig. 4 F). Taking these findings together, the TMEM192-mKeima system confirmed that TBK1 is required for lysophagy and the targeting of pTBK1 to lysosomes appears to be dependent on Ca²⁺ efflux from damaged lysosomes.

### E2 ubiquitin-conjugating enzymes UBE2L3 and UBE2N are involved in lysophagy

The ubiquitination of proteins on the damaged lysosomal membrane is an important signal to initiate lysophagy. Although several E3 ubiquitin ligases such as SCF^FBXO27, TRIM16, LRSAM1, and CUL4A have been identified to function in lysophagy, only one ubiquitin-conjugating enzyme (E2), UBE2QL1, is known to be involved in it (Huett et al., 2012; Teranishi et al., 2022; Yoshida et al., 2017; Chauhan et al., 2016; Koerver et al., 2019). Considering the case of mitophagy in which multiple E2 and E3 enzymes are involved, there may be other E2 enzymes functioning in lysophagy (Geisler et al., 2014). Therefore, we sought E2 enzymes that function in lysophagy. Previously, we performed proteomic analysis using transfection reagent-coated polystyrene beads, which cause endosomal/lysosomal membrane damage. Via this approach, we successfully identified CUL4A protein complex as a regulator of lysophagy (Teranishi et al., 2022). To explore E2 enzymes that drive lysophagy, we tested two candidates from the proteomic analysis: UBE2L3 and UBE2N.

We confirmed the effects of UBE2L3 and UBE2N deletion on lysophagy. Lysophagy was markedly reduced in UBE2L3- and UBE2N-knockdown cells compared with the level in control cells (Fig. 5 A). We then analyzed the localization of UBE2L3 and UBE2N. Cells stably expressing GFP-UBE2L3 or GFP-UBE2N were treated with LLOMe for 2 h. In untreated cells, both E2 enzymes were distributed in the cytosol, but translocated to

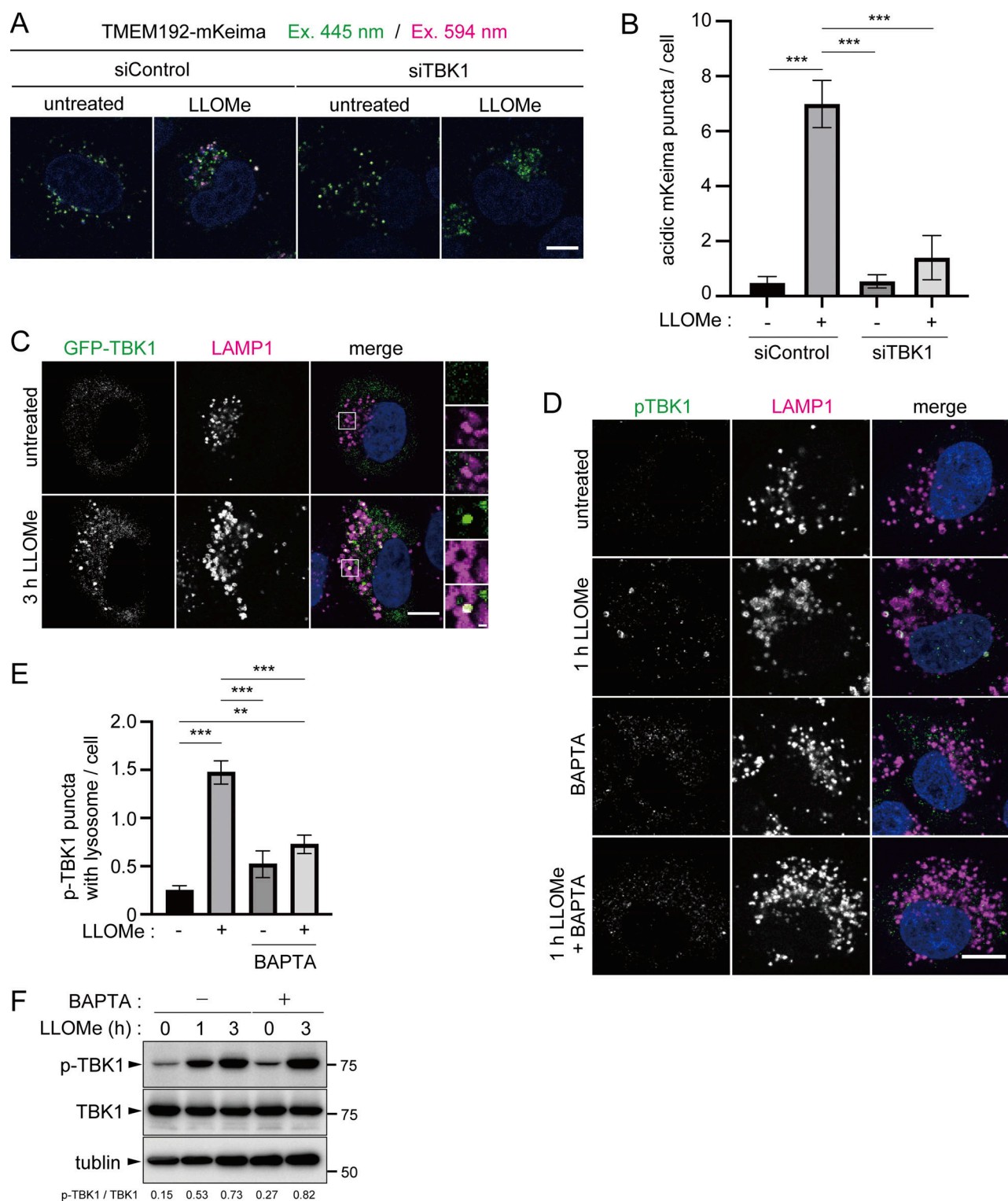

Figure 4. **TBK1 is important for lysophagy. (A)** HeLa cells stably expressing TMEM192-mKeima were treated with siRNA against TBK1 or treated with control siRNA. After 48 h of siRNA treatment, cells were cultured for 1 h in the presence of 1 mM LLOMe. After LLOMe was removed, cells were cultured in DMEM containing Hoechst for 10 h and observed by fluorescence microscopy. Scale bars, 10 μm. **(B)** Quantification of the number of acidic TMEM192-Keima puncta per cell for Fig. 4 A. >30 cells were analyzed per condition in each experiment. Error bars indicate means ± SEM in at least three independent experiments (n = 3). ***P < 0.001 (one-way ANOVA followed by Dunnett's test). **(C)** HeLa cells transiently expressing GFP-TBK1 were cultured for 3 h with or without 1 mM LLOMe, immunostained with LAMP1 antibody, and then analyzed by fluorescence microscopy. Scale bars, 10 μm. **(D)** HeLa cells were treated with or without 1 mM LLOMe and 10 μM BAPTA-AM for 1 h, immunostained with p-TBK1 and LAMP1 antibodies, and then analyzed by fluorescence microscopy. Scale bars, 10 μm. **(E)** Quantification of the number of p-TBK foci exhibiting a LAMP1 signal for Fig. 4 D. >30 cells were analyzed per condition in each experiment. Error bars indicate means ± SEM in at least three independent experiments (n = 3). ***P < 0.001 (one-way ANOVA followed by Dunnett's test). **(F)** HeLa cells were treated with LLOMe for the indicated times. Cell lysates were analyzed by immunoblotting using the indicated antibodies. Source data are available for this figure: SourceData F4.

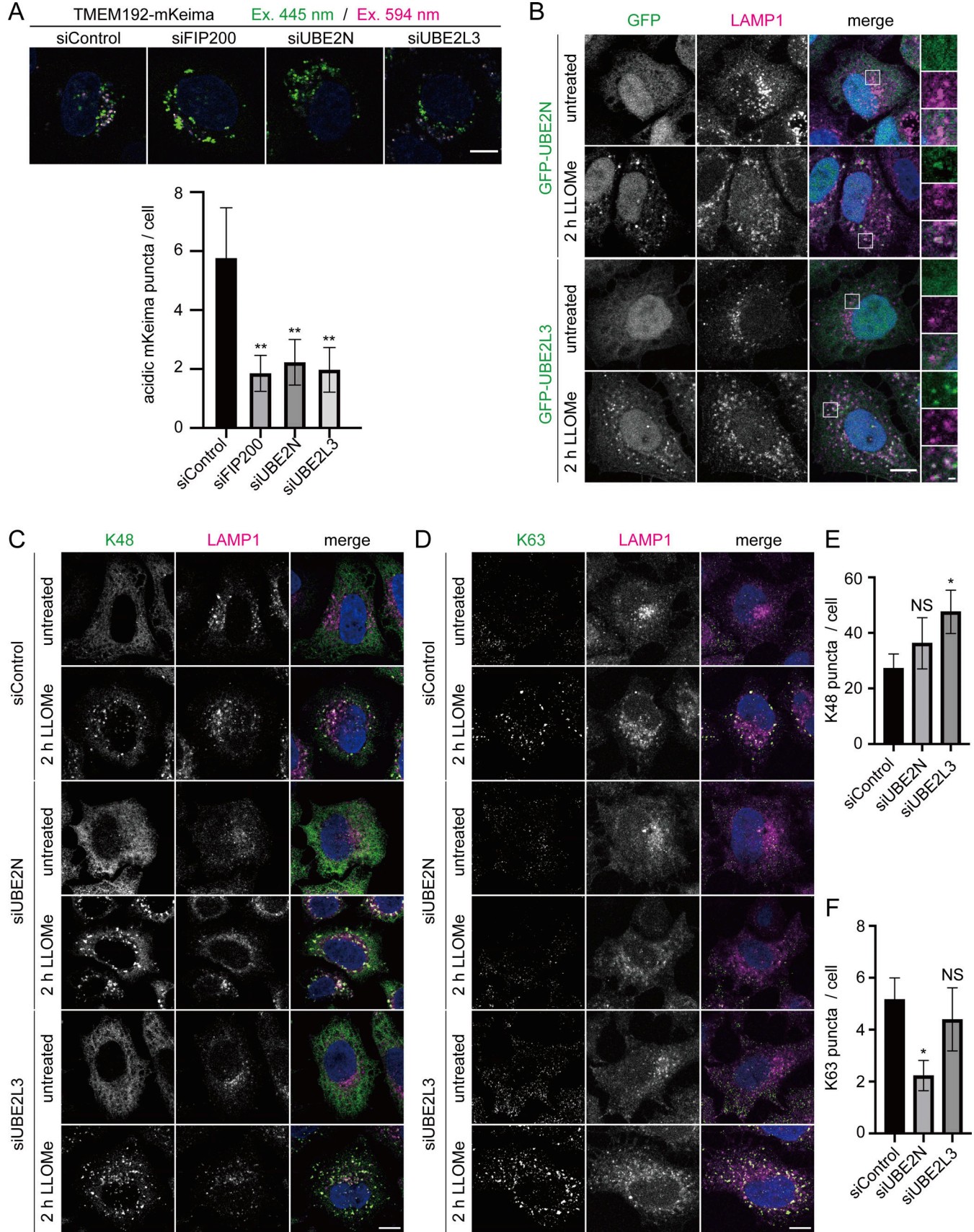

Figure 5. **UBE2N and UBE2L3 are identified as factors required for lysophagy. (A)** HeLa cells stably expressing TMEM192-mKeima were treated with siRNA against FIP200, UBE2L3, or UBE2N, or treated with control siRNA. After 48 h of siRNA treatment, cells were cultured for 6 h in the presence of 1 mM

LLOMe in DMEM containing Hoechst and observed by fluorescence microscopy. Scale bars, 10 μm. The graph shows the number of acidic TMEM192-mKeima puncta per cell in each cell. >30 cells were analyzed per condition in each experiment. Error bars indicate means ± SEM in at least three independent experiments ($n = 3$). *P < 0.05, ***P < 0.001 (one-way ANOVA followed by Dunnett's test). **(B)** HeLa cells stably expressing GFP-UBE2L3 or GFP-UBE2N were treated with or without 1 mM LLOMe for 2 h, immunostained with LAMP1 antibody, and then analyzed by fluorescence microscopy. Scale bars, 10 μm. **(C and D)** HeLa cells were treated with siRNA against each of UBE2L3, UBE2N, or control siRNA. After 48 h of siRNA treatment, cells were treated with or without 1 mM LLOMe for 2 h, immunostained with LAMP1 and polyubiquitin chain (K48 or K63) antibody, and then analyzed by fluorescence microscopy. Scale bars, 10 μm. **(E and F)** Quantification of the number of K48 or K63 puncta per cell for Fig. 5, C and D. >30 cells were analyzed per condition in each experiment. Error bars indicate means ± SEM in at least three independent experiments ($n = 3$). *P < 0.05, NS: not significant (one-way ANOVA followed by Dunnett's test).

lysosomes upon treatment with LLOMe (Fig. 5 B). We then investigated whether these E2 enzymes were required for lysosome ubiquitination. Because UBE2N catalyzes the synthesis of K63-linked polyubiquitin chains and K48- and K63-linked polyubiquitination is known to be involved in lysophagy, both K48 and K63 chains were tested with chain-specific antibodies. The results showed that UBE2L3 knockdown markedly increased the number of K48 chains on lysosomes, but did not affect the signals for K63 chains (Fig. 5, C–F). Meanwhile, UBE2N knockdown strongly reduced the number of K63 chains without affecting K48 chains (Fig. 5, C–F). Taking these findings together, we identified UBE2L3 and UBE2N as being required for lysophagy and somehow involved in K48-linked and K63-linked ubiquitination of lysosomes, respectively.

### Small-scale screening using TMEM192-mKeima probe identified TRIM10, TRIM16, and TRIM27 as regulators of lysophagy

To demonstrate whether the TMEM192-mKeima probe can be used to screen for proteins involved in lysophagy, we lastly performed a small-scale siRNA-based screen. TRIpartite Motif (TRIM) proteins are a large family of E3 ubiquitin ligases that are known to play pivotal roles in autophagy. There are >70 TRIM proteins in humans (Di Rienzo et al., 2020; Hatakeyama, 2017). Screens of TRIM proteins have been conducted using LC3 or ubiquitin as an indicator, and a number of TRIMs have been identified to function in selective autophagy (Ji et al., 2019; Mandell et al., 2014). Among them, TRIM16 was the only TRIM protein known to regulate lysophagy (Chauhan et al., 2016). We attempted to identify TRIM proteins that play a role in lysophagy via a small-scale screen with TMEM192-mKeima.

TRIM proteins are classified into 11 subfamilies based on the structural diversity of their carboxyl terminus and are known to function by forming homodimers or heterodimers with other TRIMs (Di Rienzo et al., 2020; Hatakeyama, 2017). For the screen, we picked up representative TRIM proteins from each subfamily including TRIM16; a total of 41 TRIM proteins were tested. We identified TRIM10 and TRIM27 as potential regulators of lysophagy (Fig. 6 A). Clear suppression of lysophagy was also observed by TRIM16 knockdown, confirming that this screen worked well. We followed this up by using other individual siRNAs for TRIM10, TRIM16, and TRIM27. Lysophagy was partially suppressed by knocking down TRIM10 and was markedly impaired by TRIM27 and TRIM16 knockdown (Fig. 6 B). Next, we assessed the effect of the knockdown of these TRIMs on the ubiquitination of damaged lysosomes. The signals of ubiquitin chains on lysosomes in response to damage were observed in TRIM10-, TRIM16-, and TRIM27-knockdown cells to

the same extent as in control cells (Fig. 6, C and D). These findings suggest that TRIM10 and TRIM27 are required for lysophagy but may play a role other than in the ubiquitination of damaged lysosomes.

## Discussion

In this study, we developed a novel assay specific for lysophagy using TMEM192-mKeima. By comparing the TMEM192-mKeima system with the Gal-3 clearance assay, we demonstrated that the TMEM192-mKeima assay is more specific for lysophagy than the Gal-3 clearance assay (Fig. 2). Indeed, the Gal-3 assay monitors multiple pathways in the lysosomal damage response, such as lysophagy, the ESCRT pathway, and the TFEB pathway (Fig. S4 C). These results emphasize the advantage of the specificity of TMEM192-mKeima to lysophagy, while the conventional Gal-3 assay is useful for assessing the lysosomal damage response as a whole.

Using the TMEM192-mKeima probe, we sought to elucidate the mechanism involved in the initial steps of lysophagy. We reevaluated autophagy adaptors and TBK1 to confirm their involvement in lysophagy and showed that OPTN, NDP52, and TAX1BP1 were required for lysophagy, and that TBK1 was targeted to lysosomes in response to calcium signals. We investigated further upstream of the adaptors and identified two E2 enzymes, UBE2L3 and UBE2N, as regulators of lysophagy. UBE2L3 and UBE2N were shown to be involved in lysophagy and ubiquitination of lysosomes, although their function in lysophagy is not clear yet. Furthermore, we successfully performed the small-scale screening of TRIM proteins and found that TRIM10, 16, and 27 were involved in lysophagy. Altogether, our study demonstrated that the TMEM192-mKeima probe is a useful tool for investigating lysophagy and provided new insights into the mechanism underlying lysophagy.

We demonstrated that TMEM192-mKeima evaluated lysophagy separately from other lysosomal damage responses. This advantage becomes more important as the existence of multiple pathways in lysosomal repair becomes apparent. A previous study showed that the suppression of TFEB impairs the clearance of Gal-3 puncta (Nakamura et al., 2020; Fig. 2, C and D). TFEB is known to be a master regulator of lysosomal biogenesis and autophagy at the transcriptional level (Settembre et al., 2011; Sardiello et al., 2009). Therefore, TFEB was thought to contribute to driving lysophagy by regulating the expression of ATG genes. However, the TMEM192-mKeima assay suggested that TFEB functions in the lysosome damage response, but not through lysophagy, although we cannot rule out the possibility that the TMEM192-mKeima assay may not

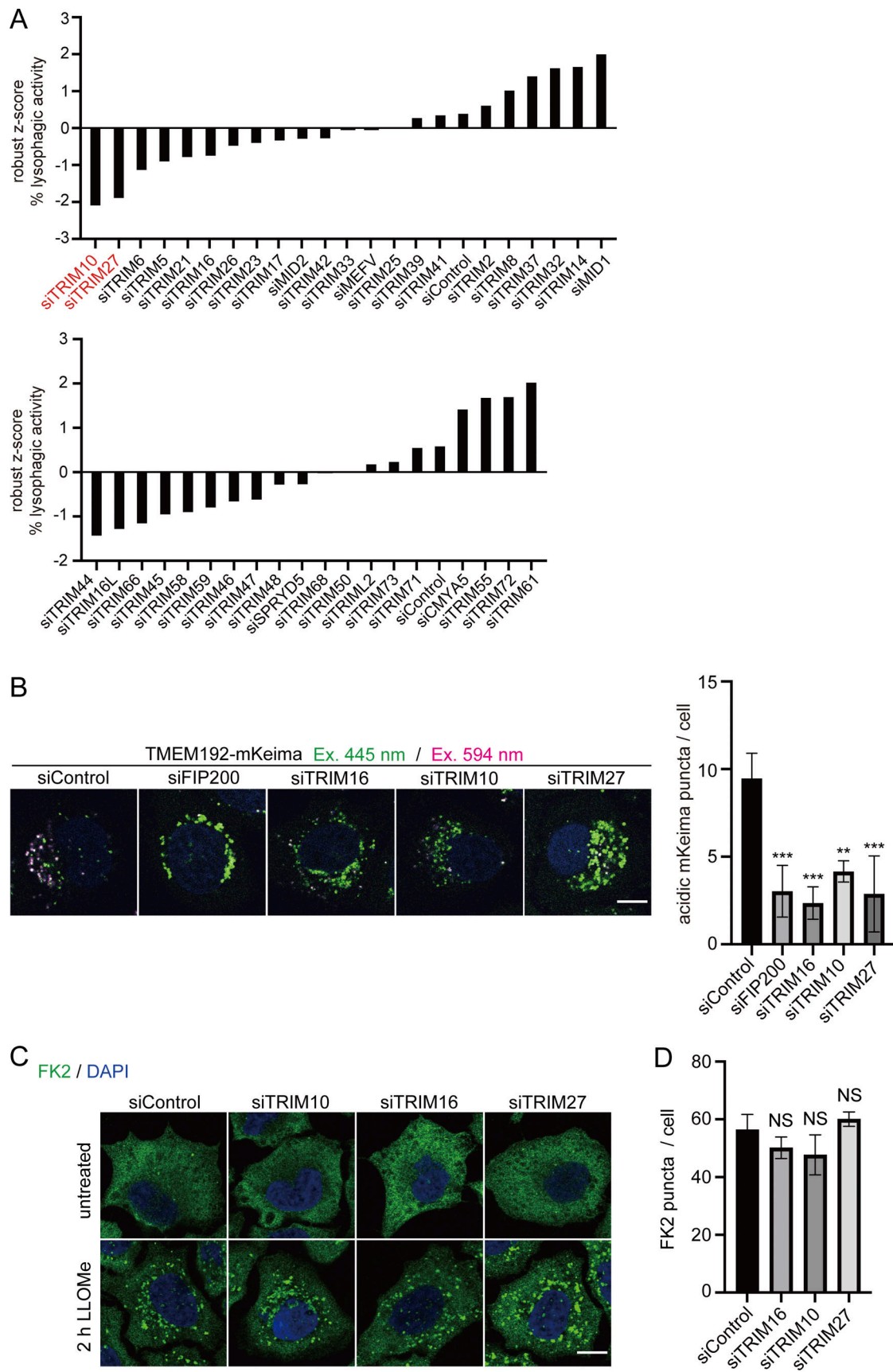

Figure 6. **Small-scaled screening using TMEM192-mKeima system identified TRIM10 and TRIM27 as factors that function in lysophagy. (A)** HeLa cells stably expressing TMEM192-mKeima were treated with siRNA against each TRIMs or treated with control siRNA. After 48 h of siRNA treatment, cells were

cultured for 6 h in the presence of 1 mM LLOMe in DMEM containing Hoechst and observed by fluorescence microscopy. Scale bars, 10 µm. The graph shows lysophagic activity with reference to the wild type. >30 cells were analyzed per condition in each experiment. For lysophagic activity in each siRNA-treated cell, a robust z-score was calculated for the overall lysophagic activity value. Red highlighting indicates those with significantly lower (less than −1.8) z-score values compared with other factors. **(B)** HeLa cells stably expressing TMEM192-mKeima were treated with siRNA against each of FIP200, TRIM16, TRIM10, TRIM27, or control siRNA. After 48 h of siRNA treatment, cells were cultured for 6 h in the presence of 1 mM LLOMe in DMEM-containing Hoechst and observed by fluorescence microscopy. The graph shows the number of acidic TMEM192-mKeima puncta per cell in each cell. >30 cells were analyzed per condition in each experiment. Error bars indicate means ± SEM in at least three independent experiments ($n = 3$). *$P < 0.05$; ***$P < 0.001$ (one-way ANOVA followed by Dunnett's test). **(C)** HeLa cells were treated with siRNA against each of TRIM10, TRIM16, TRIM27, or control siRNA. After 48 h of siRNA treatment, cells were treated with or without 1 mM LLOMe for 2 h, immunostained with polyubiquitin chain (FK2) antibody, and then analyzed by fluorescence microscopy. Scale bars, 10 µm. **(D)** Quantification of the number of polyubiquitin chain (FK2) puncta per cell for Fig. 6 C. More than 30 cells were analyzed per condition in each experiment. Error bars indicate means ± SEM in at least three independent experiments ($n = 3$). NS: not significant (one-way ANOVA followed by Dunnett's test).

evaluate the effect on lysophagy by the deletion of TFEB because of its low sensitivity and dynamic range. Our data at least highlight the concern that it might be risky to evaluate lysophagy by the Gal-3 assay alone.

There are also conflicting findings regarding the involvement of p62 in lysophagy. The Gal-3 clearance assay showed a delay in recovery from lysosomal damage in p62-knockdown cells in a previous study and in our own work (Papadopoulos et al., 2017; Fig. 3 C). Another study showed that p62 phase condensates on damaged lysosomes are required for lysophagy using LC3 as a marker (Gallagher and Holzbaur, 2023). In contrast, using Keima-LGALS3, Harper's group showed that p62 is not involved in lysophagy (Eapen et al., 2021). Consistent with this, the TMEM192-mKeima probe showed that the deletion of p62 did not affect lysophagy (Fig. 3 B), suggesting that p62 functions in the responses to lysosomal damage other than lysophagy.

Using TMEM192-mKeima, we found that the autophagy adaptor proteins OPTN, NDP52, and TAX1BP1 were involved in lysophagy (Fig. 3, D–F), although all five autophagy adaptors that we examined were recruited to damaged lysosomes (Fig. S5, A and B). We also showed that TBK1 is required for lysophagy and the recruitment of p-TBK1 to lysosomes during lysophagy was dependent on calcium signaling (Fig. 4, D and E). These results suggest that the clearance of damaged lysosomes and that of mitochondria involve similar underlying molecular mechanisms.

Taking advantage of TMEM192-mKeima, we have successfully identified E2 enzymes, UBE2L3 and UBE2N, as lysophagy regulators. Both enzymes have been reported to be involved in mitophagy (Geisler et al., 2014; Hayashida et al., 2023). UBE2L3 mainly reacts with HECT or RBR (RING-in-between-RING) E3 ligases such as Parkin. UBE2N is known to catalyze the formation of K63-linked chains on depolarized mitochondria (Geisler et al., 2014; Ordureau et al., 2014; VanDemark et al., 2001). Lysophagy uses a unique set of E3 enzymes as reported elsewhere, whereas E2 enzymes appear to be common to mitophagy.

Finally, we performed a siRNA-based screen to identify lysophagy regulators using TMEM192-mKeima and identified TRIM10, TRIM16, and TRIM27 (Fig. 6, A and B). TRIM16 has been reported to be involved in lysophagy, confirming that our approach worked well. To the best of our knowledge, TRIM10 is here reported to be involved in autophagy for the first time. As for TRIM27, conflicting findings have been obtained regarding its role in autophagy. TRIM27 has been shown to be a negative regulator of autophagy by promoting the proteasomal degradation of

UKL1 (Yang et al., 2022). Meanwhile, another study showed that TRIM27 facilitates mitophagy via TBK1 (Garcia-Garcia et al., 2023). Our findings are more consistent with this latter report. Despite TRIM10 and TRIM27 being E3 enzymes, we did not observe changes in the ubiquitination of lysosomes by their deletion (Fig. 6, C and D). Further studies are needed to elucidate their roles in autophagy.

Currently, several Gal-3-based probes are in use for monitoring lysophagy: GFP-Gal-3, mRFP-GFP-Gal-3, and Keima-LGALS3 (Eapen et al., 2021; Maejima et al., 2013). The difference between probes using Gal-3 versus TMEM192 is that Gal-3 is a cytosolic and secretory protein, whereas TMEM192 is a transmembrane protein of lysosomes. Caution should be taken when using both of these markers in certain situations. For example, RFP-EGFP-Gal3 in the cytosol and extracellular space can be delivered to lysosomes through basal non-selective autophagy and endocytosis, respectively, which leads to the presence of RFP-only puncta under basal conditions, as described in the paper by Harper's group (Eapen et al., 2021). The same process may occur with Keima-LGALS3. Meanwhile, TMEM192-mKeima labels the cargo itself, as is the case with probes for mitophagy and ERphagy. In the current study, however, a few weakly lysophagic TMEM192-mKeima puncta were observed under basal conditions, which did not disappear in FIP200-knockdown cells (Fig. 1 E), indicating that they may represent invagination of the lysosomal membrane into the lysosome through microautophagy or MBV formation. Therefore, when using these probes, their specific characteristics should be taken into consideration.

One concern regarding the TMEM192-mKeima assay is that it has a poor dynamic range compared with the Gal-3 clearance assay. As demonstrated in Fig. S4 C, lysophagy is only one of the several parallel pathways monitored by the Gal-3 assay. It appears that lysophagy, Alix, and TFEB pathways contribute to the recovery of damaged lysosomes to the same degree and other pathways may also be involved (Fig. S4 C). Therefore, the small number of acidic TMEM192-mKeima compared with Gal-3 dots might be reasonable, and this may limit the utility of the assay.

Since our discovery of lysophagy, several lysosome damage responses have been revealed in the last decade: the ESCRT pathway for membrane repair, the TFEB pathway for lysosomal biogenesis, the annexin pathway for calcium-responding membrane repair, sphingomyelin exposure on damaged lysosomes, and lipid transport via ER–lysosome contact sites (Maejima et al., 2013; Skowyra et al., 2018; Papadopoulos et al.,

2020; Tan and Finkel, 2022; Radulovic et al., 2022; Yim et al., 2022; Niekamp et al., 2022). In cells, these responses to lysosomal damage may be coordinated to cope with the stress of lysosomal damage, but the crosstalk between them is unclear. Further studies are needed to elucidate the mechanism of lysophagy and to obtain a comprehensive understanding of the responses to lysosomal damage. The TMEM192-mKeima system will be very useful in achieving these goals.

## Materials and methods

### Cell culture and transfection

All cell lines (HeLa Kyoto, U2OS, and Plat-E cells) were cultured at 37°C with 5% $CO_2$ in DMEM (Sigma-Aldrich) supplemented with 10% fetal bovine serum and the appropriate antibiotics. Plat-E cells were a generous gift from Dr. Kitamura (The University of Tokyo, Japan). U2OS cells were from laboratory stock (Oe et al., 2022). HeLa cells with a knockout of autophagy-related genes (ATGs) were generated previously (Nakamura et al., 2020). Briefly, cells were transfected with pX458-encoding sgRNA targeting ATGs using ViaFect (Promega). 24 h after transfection, GFP-positive cells were isolated using a cell sorter (SH800ZFP; SONY) and single clones were obtained. Lipofectamine 2000 (Invitrogen) was used for transient transfection and cells were used 24 h after transfection for immunofluorescence.

### Plasmids and virus production

The plasmids encoding mKeima (plasmid #54597; Addgene) and TMEM192 (plasmid #102930; Addgene) were purchased from Addgene. The pMRX-IRES-puro and pMRX-IRES-bsr vectors were gifts from Dr. Yamaoka (Tokyo Medical and Dental University, Tokyo, Japan). For the preparation of retroviruses, a DNA fragment of TMEM192-mKeima was inserted into pMRX-IRES-puro and pMRX-IRES-bsr vectors (Saitoh et al., 2003). Plat-E cells were transfected with the retroviral plasmid together with pCMV-VSV-G using PEI and the virus was collected from the supernatant (Saitoh et al., 2003). Cells were infected with retrovirus and stable transformants were selected with 1 µg/ml puromycin or 5 µg/ml blasticidin. DNA fragments encoding p62, OPTN, NDP52, and UBE2L3 were amplified from HeLa Kyoto cDNA. DNA fragments encoding other receptors were amplified from plasmids encoding TAX1BP1, NBR1 (gift from Dr. Nakagawa), and UBE2N (from plasmid #12460; Addgene). These fragments were inserted into pENTR1A (Invitrogen). pENTR1A-autophagy adapters, TBK1, UBE2L3, and UBE2N were transferred into pMRX-IRES-puro-GFP, pMRX-IRES-puro-FLAG, and pcDNA3.1-GFP using the Gateway system (Invitrogen). Primers used in this study are listed below.

p62_Fw; 5′-ACTGGTCGACATGGCGTCGCTCACCGTGAA-3′
p62_Rv; 5′-ACTGCTCGAGTCACAACGGCGGGGGATGC-3′
OPTN_Fw; 5′-ACTGGGATCCGGATGTCCCATCAACCTCTCAG-3′
OPTN_Rv; 5′-ACTGCTCGAGTTAAATGATGCAATCCATCA-3′
NDP52_Fw; 5′-ACTGGGATCCGGATGGAGGAGACCATCAAAGA-3′
NDP52_Rv; 5′-ACTGGATATCTCAGAGAGAGTGGCAGAACA-3′
UBE2L3_Fw; 5′-TCAGTCGACTGGATCATGGCGGCCAGCAGGAGG-3′

UBE2L3_Rv; 5′-ATCTCGAGTGCGGCCTTAGTCCACAGGTCGCTTT-3′
TAX1BP1_Fw; 5′-ACTGGGATCCGGATGACATCCTTTCAAGAAG-3′
TAX1BP1_Rv; 5′-ACTGCTCGAGCTAGTCAAAATTTAGAACATTC-3′
NBR1_Fw; 5′-TCAGTCGACTGGATCATGGAACCACAGGTTACTC-3′
NBR1_Rv; 5′-ATCTCGAGTGCGGCCTCAATAGCGTTGGCTGTAC-3′
UBE2N_Fw; 5′-TCAGTCGACTGGATCATGGCCGGGCTGCCCCGC-3′
UBE2N_Rv; 5′-ATCTCGAGTGCGGCCTTAAATATTATTCATGGC ATATAG-3′.

### siRNA knockdown

siRNAs against OPTN (Hs01_00149664 and Hs01_00149665), CALCOCO2 (Hs01_00137944 and Hs01_00137945), TAX1BP1 (Hs02_00313970 and Hs02_00313971), TBK1 (Hs01_00062067), UBE2L3 (Hs01_00049991), UBE2N (Hs01_00012964), TRIM10 (Hs02_00342285), TRIM16 (Hs02_00341641), and TRIM27 (Hs02_00341721) were purchased from Sigma-Aldrich. FIP200 (5′-GAUCUUAUGUAGUCGUCCA-3′), ATG14 (5′-GCAAGAUGAGGA UUGAACA-3′), ATG7 (5′-GGAGUCACAGCUCUUCCUU-3′), ATG12 (5′-GAACACCAAGUUUCACUGU-3′), ATG16 (5′-GAUCUUAUGUAG UCGUCCA-3′), Stx17 (5′-AATTAAGTCCGCTTCTAAGGTTTCC-3′), and ALIX (5′-CCUGGAUAAUGAUGAAGGA-3′) siRNAs were constructed by Sigma-Aldrich. P62 (s16960), NBR1 (s227478), and TRIMs siRNAs for mini screening (MID2 [s21756], TRIM2 [s533397], TRIM5 [s39922], TRIM6 [s42186], TRIM8 [s37682], TRIM10 [s19668], TRIM14 [s19017], TRIM16 [s20874], TRIM17 [s27504], MID1 [s8779], MEFV [s502557], TRIM21 [s13462], TRIM23 [s1544], TRIM25 [s15204], TIRM26 [s15204], TRIM27 [s11961], TRIM32 [s22741], TRIM33 [s28372], TRIM37 [s9083], TRIM39 [s32222], TRIM41 [s40531], TRIM42 [s50216], TRIM44 [s29408], TRIM45 [s37139], TRIM48 [s35580], TRIM50 [s43948], SPRYD5 [s39417], TRIM55 [s39300], TRIM58 [s24690], TRIM59 [s50206], TRIM61 [s53003], TRIM66 [s458620], TRIM68 [s30235], TRIM16L [s44968], TRIM71 [s43599], TRIM72 [s54617], TRIM73 [s51790], CMYA5 [s47440] and TRIML2 [s47570]) were purchased from Thermo. For TFEB knockdown, ON-TARGET plus human TFEB (L-009798-00-0005) siRNA (Dharmacon) was used. siRNA was introduced into HeLa cells using Lipofectamine RNAiMAX (Invitrogen) and the transfected cells were used for the subsequent experiment after 48 or 72 h. The silencing efficacy was assessed by Western blotting or quantitative PCR.

### Antibodies and reagents

For immunoblotting and immunofluorescence, the following antibodies were used in this study: anti-FLAG (mouse; Sigma-AldrichF1804; Sigma-Aldrich), anti-α-tubulin (mouse; B512; Sigma-Aldrich), anti-TBK1 (rabbit; EP611Y; Abcam), anti-p-TBK1 (Ser172; rabbit; D52C2; Cell Signaling Technology), anti-LAMP1 (mouse; sc-19992; Santa Cruz Biotechnology), anti-p62 (rabbit; PM045; MBL), anti-ubiquitin antibody, Lys48-specific (rabbit; 05-1307; Millipore), anti-ubiquitin antibody, Lys63-specific (rabbit; 05-1308; Millipore), and anti-polyubiquitin antibody FK2 (mouse; 0918-2; Nippon Bio-Test Laboratories). The secondary antibodies used for immunoblotting were horseradish peroxidase (HRP)-conjugated goat anti-rabbit IgG (111-035-003; Jackson ImmunoResearch) and HRP-conjugated goat anti-mouse IgG (115-035-003; Jackson ImmunoResearch). The secondary antibodies used for immunofluorescence were goat anti-rabbit

Alexa Fluor 488 preabsorbed (ab1500085; Abcam) and goat anti-mouse IgG (H+L) cross-adsorbed Alexa Fluor 568 (A11004; Invitrogen). LLOMe was purchased from Sigma-Aldrich. Depending on the experiment, cells were treated for 1 h with 10 µM BAPTA-AM (Dojindo) or for 20 h with 250 nM bafilomycin A1 (Cayman Chemical).

## Fluorescence microscopy
Cells were cultured on coverslips and then fixed with 4% paraformaldehyde in phosphate-buffered saline (PBS) for 15 min. They were washed twice with PBS, permeabilized with PBS containing 50 µg/ml digitonin, blocked with PBS containing 0.2% gelatin, and then incubated with each primary antibody. After washing with PBS three times and treatment with a secondary antibody, the cells were mounted onto glass slides with ProLong Diamond (Thermo Fisher Scientific). The cells were examined using a 60× 1.3 NA oil immersion objective and an FV3000 confocal microscope (Olympus) operated using FV31S-SW (version 2.3.1.163) at room temperature. The images were adjusted using Fiji (National Institutes of Health; https://imagej.net/Fiji).

## Lysophagy assay
TMEM192-mKeima expressing cells were precultured overnight in the aforementioned DMEM on a glass-bottomed dish (D11130H; Matsunami Glass). To induce lysophagy, cells were treated with 1 mM LLOMe, followed by culture in DMEM (040-30095; Fujifilm)-containing Hoechst 33342 (H342; Dojindo) without LLOMe. After incubation, the cells were subjected to fluorescence microscopy. The cells were examined using a 60× 1.3 NA oil immersion objective and an FV3000 confocal microscope (Olympus) operated using FV31S-SW (version 2.3.1.163) at room temperature. The cells were observed under neutral and acidic conditions using excitation wavelengths of 440 and 594 nm, respectively. The number of puncta under 594 nm excitation was determined using Fiji (National Institutes of Health; https://imagej.net/Fiji).

First, the maximum signal at 445 nm was identified, and the intensity and position of each signal were measured. The intensity of each signal at 594 nm was then measured at the same position, and the value at 594 nm was compared with the value at 445 nm. If this ratio was above the set criteria, the signal was considered to indicate "acidic mKeima punctum." Lysophagic activity in the cell in each experiment was quantified according to the above criteria.

## Immunoblotting
Cells were washed twice with ice-cold PBS and cell lysates were prepared in a lysis buffer (50 mM Tris-HCl, pH 7.4, 150 mM NaCl, 1 mM EDTA, 1% Triton X-100) supplemented with complete EDTA-free protease inhibitor and 1 mM phenylmethylsulfonyl fluoride (PMSF). After centrifugation, supernatants were mixed with SDS sample buffer (final concentration 56 mM Tris-HCl, pH 6.8, 0.1 M DTT, 2% SDS, 6% glycerol, and 2.4% Bromophenol Blue) and heated at 95°C for 5 min. The samples were analyzed by SDS-PAGE and transferred to polyvinylidene difluoride membranes (IPVH00010; Merck Millipore). The membranes were blocked with 1% skim milk in TBST (TBS and 0.1% Tween 20) buffer and incubated overnight at 4°C with primary antibodies. The membranes were washed and then incubated for 1 h at room temperature with HRP-conjugated secondary antibodies. After washing, immunoreactive bands were detected by Luminata Forte (WBLUF0100; Merck Millipore) or ImmunoStar LD (290-69904; FUJIFILM) using a ChemiDoc Touch Imaging System (Bio-Rad).

## Magic Red assay
Magic Red (937; ImmmunoChemistry) is adjusted by dissolving in 50 µl of DMSO. After adding 1 µl of Magic Red solution to 2 ml of Hoechst 33342 containing DMEM medium and incubating at 37°C for 1 h, live cells under no treatment, 1 mM LLOMe treated, or LLOMe washout conditions were observed under a fluorescence microscope.

## Statistical analysis
GraphPad Prism8 was used for graphing and statistical analysis. Quantitative data were displayed as mean ± SEM of at least three independent experiments ($n$ = 3). Data distribution was assumed to be normal, but this was not formally tested. The significance of differences was evaluated by unpaired two-tailed Student's $t$ test or one-way ANOVA followed by Dunnett's test. *P < 0.05, **P < 0.01, and ***P < 0.001 were considered statistically significant.

## Online supplemental material
Fig. S1 shows that TMEM192-mKeima localizes to the lysosome even under conditions with lysosomal damage. Fig. S2 shows that the conversion of mKeima increases in a time- and concentration-dependent manner with LLOMe. Fig. S3 shows that conversion to acidic mKeima depends on the acidity of the lysosome and autophagy pathway. Fig. S4 shows knockdown of ALIX or TFEB does not affect the number of acidic mKeima in the TEME192-mKeima assay but causes defects in the clearance of damaged lysosomes in the GFP-Gal3 assay. Fig. S5 shows that autophagic adaptors localize to lysosomes under conditions with lysosomal damage. Fig. S6 shows that the recovery of lysosomal enzyme activity is suppressed in Penta-KO cells.

## Data availability
Data supporting the results of this study are available from the corresponding author upon reasonable request.

## Acknowledgments
We thank Ms. Miki Iwatani for technical support, Dr. Ichiro Nakagawa for providing the plasmids, and Dr. Richard Youle for providing the autophagy adaptor-deficient HeLa cells.

This work was supported by the Japan Science and Technology Agency (Core Research for Evolutional Science and Technology; JPMJCR17H6), the Japan Agency for Medical Research and Development (AMED; grant numbers JP17gm5010001, JP17gm0610005 and JP23gm1410014), and JSPS KAKENHI (grant number JP19K06637, 22H04982 and 22KJ2105). Open Access funding provided by Osaka University.

Author contributions: T. Shima, A. Kuma, and T. Yoshimori designed the project. T. Shima, A. Kuma, and R. Matsuda performed the experiments with the help of M. Ogura. T. Shima and A. Kuma wrote the manuscript. All authors analyzed and discussed the results and commented on the manuscript.

Disclosures: The authors declare no competing interests exist.

Submitted: 14 April 2022

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

# Supplemental material

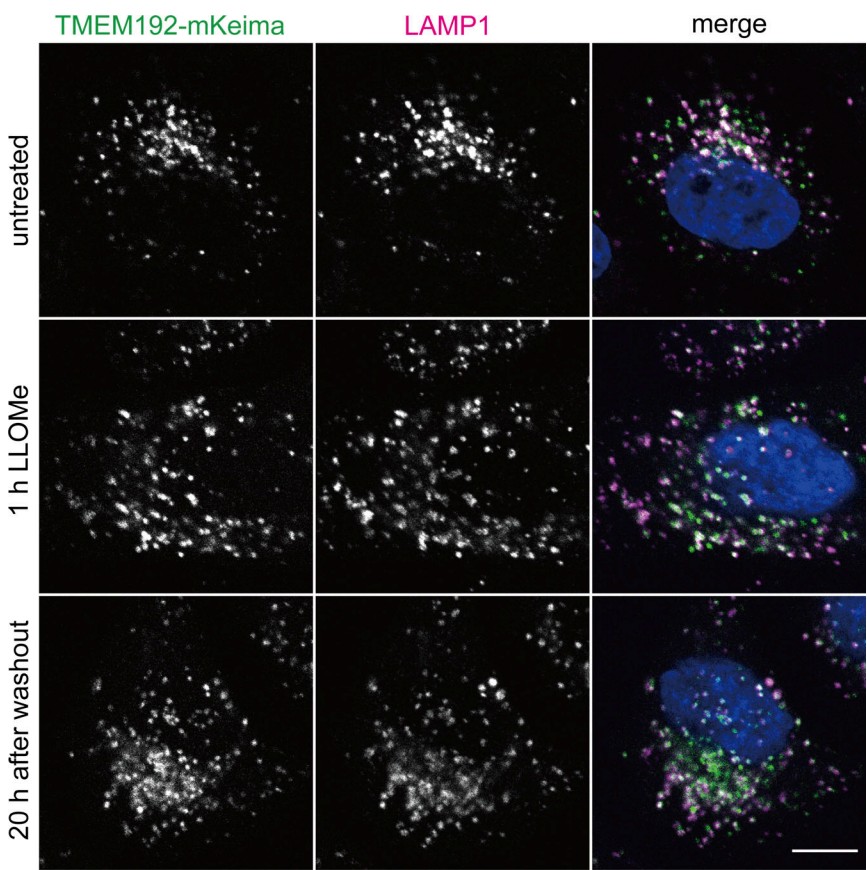

Figure S1.  **TMEM192-mKeima localizes to the lysosome even under conditions with lysosomal damage.** HeLa cells expressing TMEM192-mKeima were cultured for 1 h in the presence of 1 mM LLOMe. After LLOMe was removed, cells were cultured for 20 h in DMEM. Cells were immunostained with LAMP1 antibody and then analyzed by fluorescence microscopy. Scale bars, 10 µm.

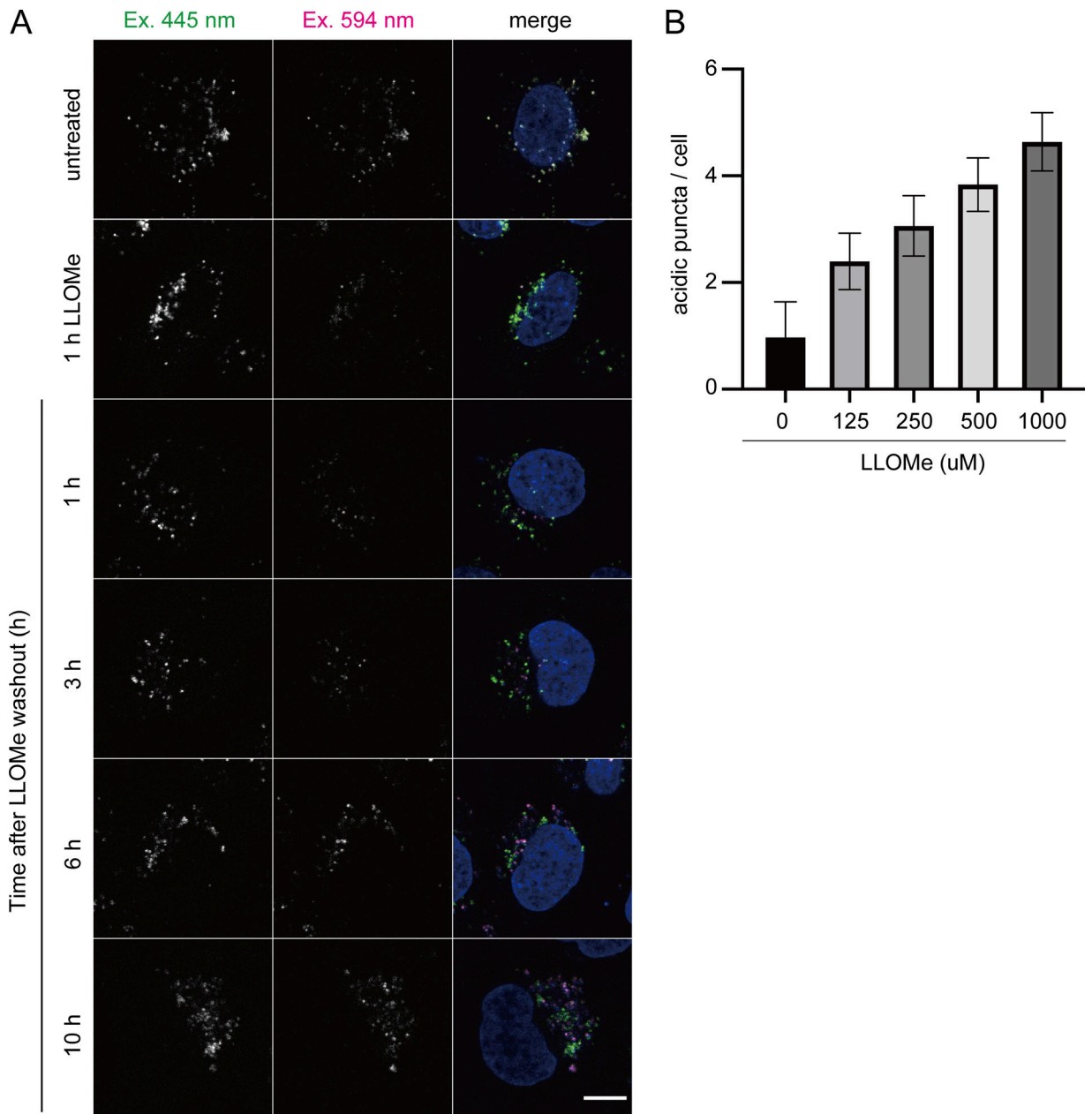

**Figure S2. The conversion of mKeima increases in a time- and concentration-dependent manner with LLOMe. (A)** HeLa cells stably expressing TMEM192-mKeima were cultured for 1 h in the presence of 1 mM LLOMe, followed by culture in DMEM containing Hoechst without LLOMe. After incubation for 1, 3, 6, or 10 h, cells were observed under a fluorescence microscope. Scale bars, 10 μm. **(B)** HeLa cells stably expressing TMEM192-mKeima were incubated for 1 h in the presence of 125, 250, 500, or 1,000 μM LLOMe. After LLOMe was removed, cells were cultured in DMEM containing Hoechst for 20 h and then analyzed by fluorescence microscopy. The graph shows the number of acidic TMEM192-mKeima puncta per cell. Error bars indicate means ± SEM in at least three independent experiments ($n$ = 3). >30 cells were analyzed per condition in each experiment.

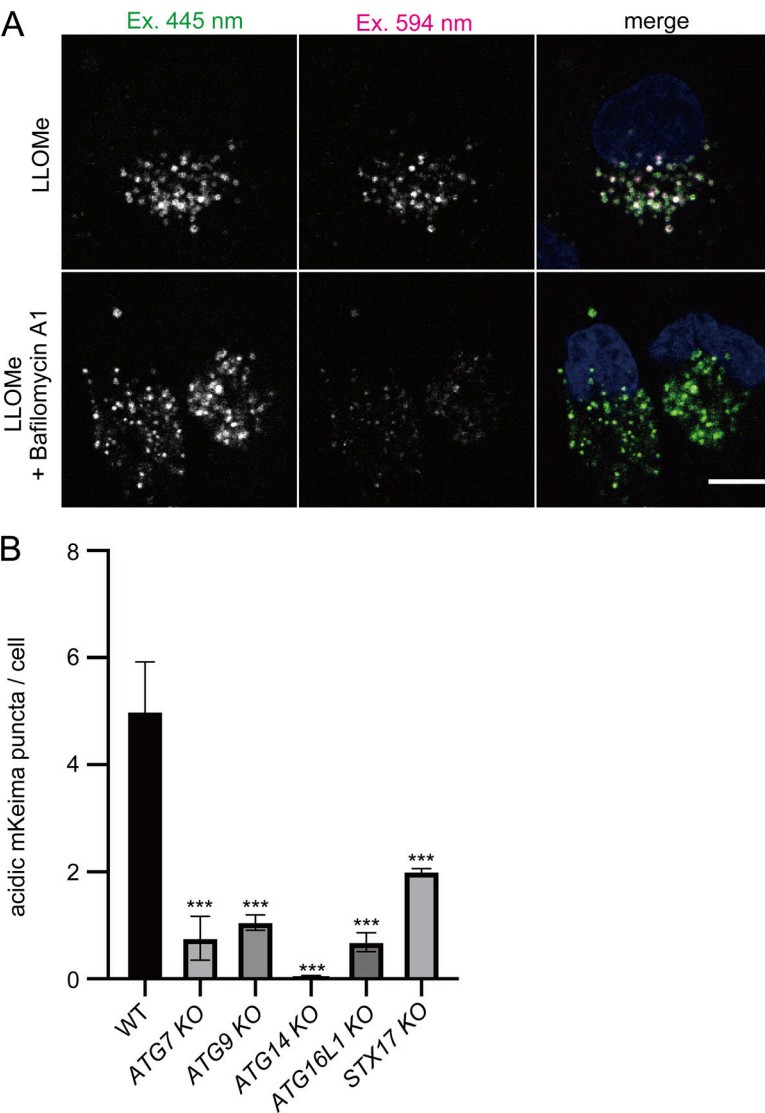

Figure S3.   **Conversion to acidic mKeima depends on the acidity of the lysosome and autophagy pathway. (A)** HeLa cells stably expressing TMEM192-mKeima were incubated for 1 h in the presence of 1 mM LLOMe. After LLOMe was removed, cells were cultured in DMEM containing Hoechst and bafilomycin A1 for 20 h and then analyzed by fluorescence microscopy. Scale bars, 10 μm. **(B)** WT, ATG7-KO, ATG9-KO, ATG14-KO, ATG16L1-KO, and STX17-KO HeLa cells expressing TMEM192-mKeima were cultured for 1 h in the presence of 1 mM LLOMe, followed by culture in DMEM containing Hoechst without LLOMe. After 21 h of incubation, cells were observed by fluorescence microscopy. The graph shows the number of acidic TMEM192-mKeima puncta per cell in WT and KO HeLa cells. >30 cells were analyzed per condition in each experiment. Error bars indicate means ± SEM in at least three independent experiments ($n = 3$). ***$P < 0.001$ (one-way ANOVA followed by Dunnett's test).

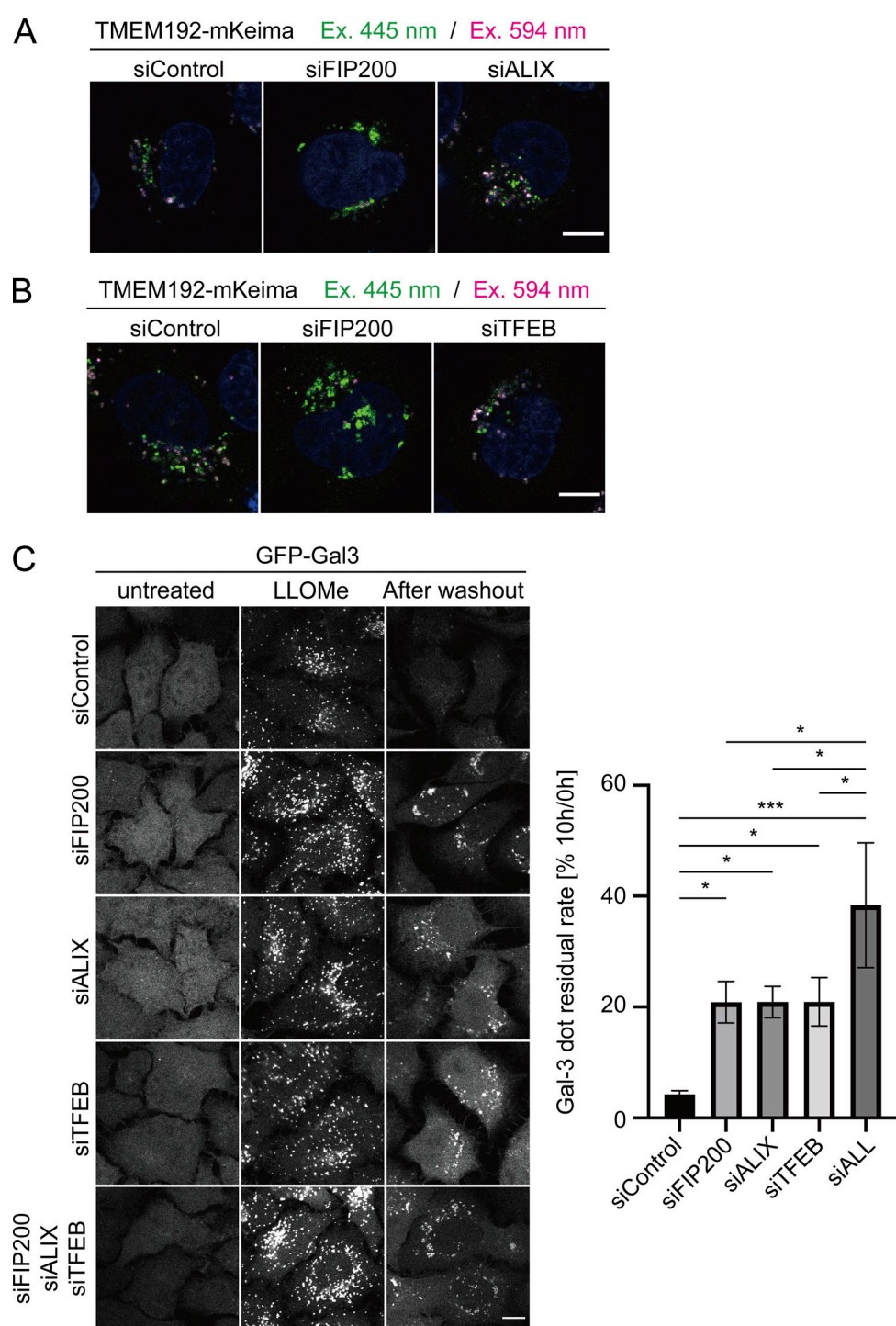

Figure S4. **Knockdown of ALIX or TFEB does not affect the number of acidic mKeima in the TEME192-mKeima assay but causes defects in the clearance of damaged lysosomes in the GFP-Gal3 assay. (A and B)** HeLa cells stably expressing TMEM192-mKeima were treated with siRNA against FIP200, Alix, or TFEB, or treated with control siRNA. After siRNA treatment, cells were cultured for 5 h in the presence of 1 mM LLOMe and Hoechst. Cells were observed by fluorescence microscopy. Scale bars, 10 μm. **(C)** HeLa cells stably expressing GFP-Gal3 were treated with siRNA against FIP200, Alix, TFEB, or treated with control siRNA. After siRNA treatment, cells were cultured for 1 h in the presence of 1 mM LLOMe, followed by culture in DMEM without LLOMe. After 10 h, cells were observed by fluorescence microscopy. Scale bars, 10 μm. The graph shows the percentage of Gal-3 puncta remaining after 10 h of incubation. >30 cells were analyzed per condition in each experiment. Error bars indicate means ± SEM in at least three independent experiments ($n = 3$). **P < 0.01 (one-way ANOVA followed by Dunnett's test).

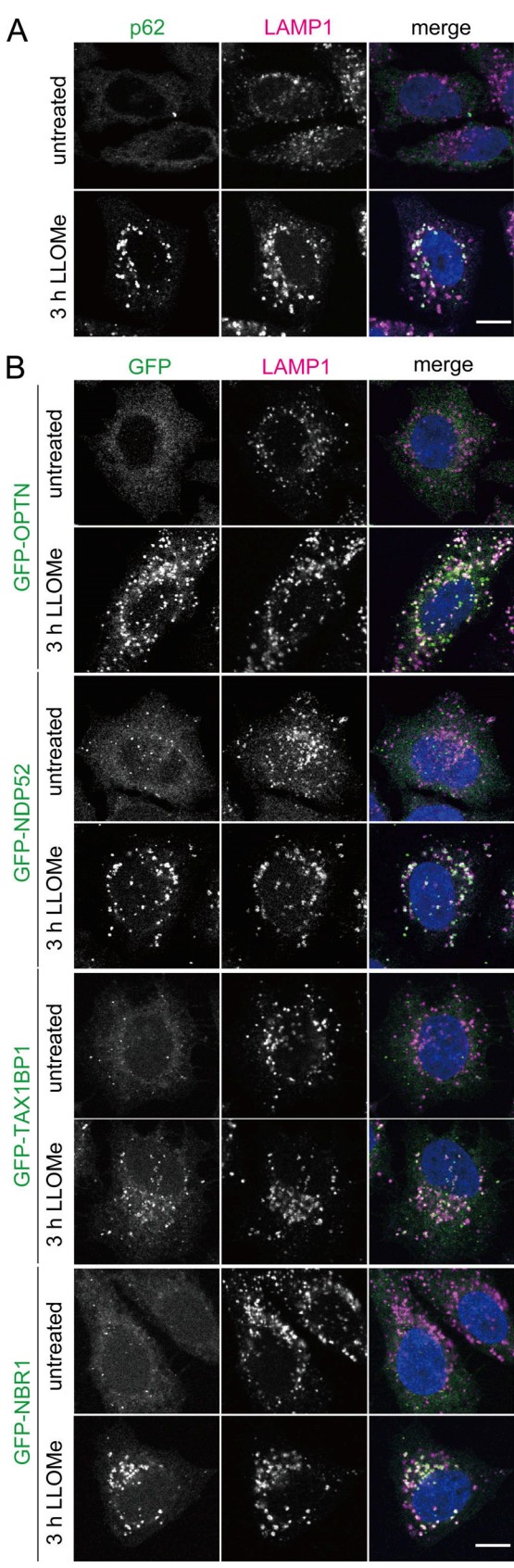

Figure S5. **Autophagic adaptors localize to lysosomes under conditions with lysosomal damage. (A)** HeLa cells were treated with or without 1 mM LLOMe for 3 h, immunostained with p62 and LAMP1 antibodies, and then analyzed by fluorescence microscopy. Scale bars, 10 µm. **(B)** HeLa cells stably expressing GFP-OPTN, GFP-NDP52, GFP-TA1BP1, or GFP-NBR1 were treated with or without 1 mM LLOMe for 3 h, immunostained with LAMP1 antibody, and then analyzed by fluorescence microscopy. Scale bars, 10 µm.

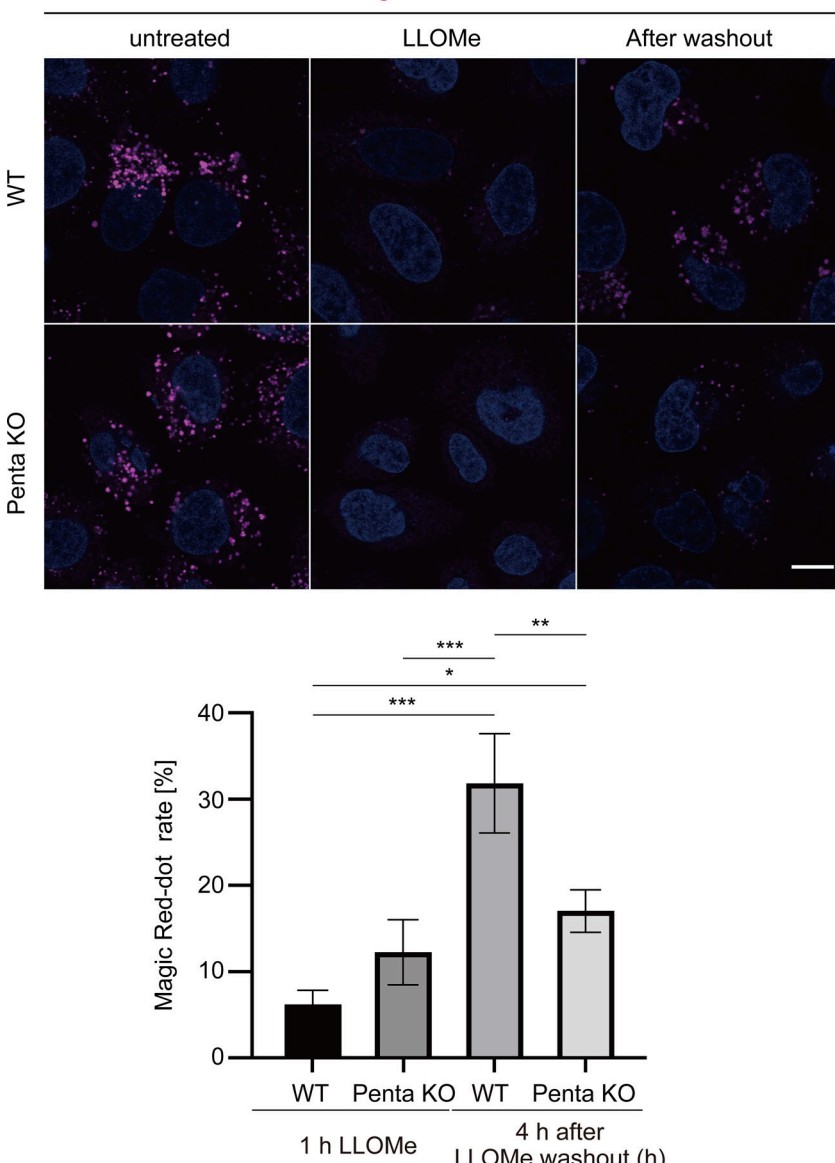

Figure S6. **The recovery of lysosomal enzyme activity is suppressed in Penta-KO cells.** WT or Penta-KO HeLa cells were cultured for 1 h in the presence of 1 mM LLOMe, followed by culture in DMEM medium containing Hoechst without LLOMe. After incubation for 4 h, cells were observed under a fluorescence microscope. Scale bars, 10 µm. The graph shows the percentage of Magic Red dots in cells after 4 h of incubation compared with untreated cells. >30 cells were analyzed per condition in each experiment. Error bars indicate means ± SEM in at least three independent experiments (*n* = 3). **P < 0.01, NS: not significant (one-way ANOVA followed by Dunnett's test).

