## [Peer Review File · The Journal of Cell Biology]

The TMEM192-mKeima probe specifically assays lysophagy and reveals its initial steps.

Takayuki Shima, Monami Ogura, Ruriko Matsuda, Shuhei Nakamura, Natsuko Jin, Tamotsu Yoshimori, and Akiko Kuma

Corresponding Author(s): Tamotsu Yoshimori, Osaka University and Akiko Kuma, Osaka University

Review Timeline:

Submission Date:	2022-04-14
Editorial Decision:	2022-06-03
Revision Received:	2023-04-28
Editorial Decision:	2023-06-04
Revision Received:	2023-08-23
Editorial Decision:	2023-08-31
Revision Received:	2023-09-08

Monitoring Editor: Xiaochen Wang

Scientific Editor: Dan Simon

Transaction Report:

DOI: <https://doi.org/10.1083/jcb.202204048>

June 3, 2022

Re: JCB manuscript #202204048

Prof. Tamotsu Yoshimori
Osaka University
2-2 Yamadaoka, Suita
Osaka 565-0871
Japan

Dear Prof. Yoshimori,

Thank you for submitting your manuscript "The novel probe TMEM192-mKeima specifically assays lysophagy and reveals its initial steps." The manuscript has been evaluated by expert reviewers, whose reports are appended below. Unfortunately, after an assessment of the reviewer feedback, our editorial decision is against publication in JCB.

You will see that the reviewers express considerable interest in a lysophagy-specific reporter but also feel that the study in its current form does not provide a substantial advance in our understanding of the factors that are necessary or not essential for lysophagy. Other major concerns are whether TMEM192-mKeima is as sensitive as currently used Gal3-based assays and need for additional time points as well as assays in other cell types to better characterize the behavior and general utility of TMEM192-mKeima.

In both their reviews and private comments, the reviewers convey the sentiment that the study regrettably does not provide the kind of cutting-edge and highly novel findings that would be appropriate for the JCB Report format. However, given interest in the topic, we would be open to resubmission to JCB of a significantly revised and expanded manuscript for the Tools format. This revision would need to fully address the reviewers' concerns by providing a thorough characterization of TMEM192-mKeima as well as directly comparing it to relevant Gal3-based probes. As proof of concept of a lysophagy-specific reporter, a revised study should provide new insights into lysophagy that are not revealed by currently used Gal3-based assays. Please note that priority and novelty would be reassessed at resubmission. If you would like to resubmit this work to JCB, please contact the journal office to discuss an appeal of this decision or you may submit an appeal directly through our manuscript submission system.

Although your manuscript is intriguing, we feel that the points raised by the reviewers are more substantial than can be addressed in a typical revision period. If you wish to expedite publication of the current data, it may be best to pursue publication at another journal. The journal office can transfer your reviewer comments to another journal upon request.

Regardless of how you choose to proceed, we hope that the comments below will prove constructive as your work progresses. We would be happy to discuss the reviewer comments further once you've had a chance to consider the points raised in this letter. You can contact the journal office with any questions, cellbio@rockefeller.edu or call (212) 327-8588.

Thank you for thinking of JCB as an appropriate place to publish your work.

Sincerely,

Xiaochen Wang, PhD
Monitoring Editor
Journal of Cell Biology

Dan Simon, PhD
Scientific Editor
Journal of Cell Biology

Reviewer #1 (Comments to the Authors (Required)):

The molecular response to permeabilized or otherwise damaged lysosomes is very topical but still poorly understood. The pathway is of great interest because it is connected to various conditions including neurodegeneration in humans and represents a potential handle for cancer treatment. A major problem in the field is the differentiation of the two branches in quantitative assessments: lysosome repair on the one hand, or lysosome degradation through lysophagy on the other hand. A commonly used assay following Gal3, or a recent improvement with Gal3-mKeima (Eapen et al., 2021), is useful for many questions but

cannot differentiate between the branches because both result in Gal3 being in an acidic environment. The authors now report on the development of cells that carry the mKeima reporter on the cytosolic tail of resident lysosomal protein TMEM192. Therefore, a fluorescence shift only occurs after autophagocytosis, but not upon repair and reacidification. The authors extensively characterize the cells providing evidence for the applicability of their approach and provide new findings and confirmations on the process of lysophagy.

The development of the cell line is elegant and very useful. The approach will be widely accepted and used in the field. Probably the most interesting finding using the approach in this manuscript is the proof of the differential timing of repair and lysophagy (Fig. 2A), which answers a long-standing question. The data on the involvement of autophagy receptors and on TBK1 is a great addition because they are conclusive albeit with somewhat limited scope because some of it was known already (most recently in Eapen et al., 2021, in *Elife*). Many extra time points, quantifications, statistics and controls are missing (see below) that need to be added before potential acceptance.

Major points:

- 1) Fig. 1B: The authors present only one time point. To characterize the cells and how the reporter behaves, they need to show images for more time points than just the 20h time point. They looked at the time course in Fig. 2A, but there the images are missing. Could a parallel stain with Gal3 be done, so to see the extent of damage. Are there any fluctuations in the signal? How does mKeima react to efflux of protons upon lysosome damage? It would also be nice to see red TMEM192-mKeima wrapped in LC3 for confirmation.
- 2) Fig. 2A: the corresponding images need to be shown. Statistic (errors and significance) should be shown. The time points before wash out would be important as reference. The entire time course should be repeated in at least one ATG-KO condition.
- 3) Fig. 2B-G: The authors compare effects on GFP-Gal3 and TMEM192-mKEIMA, but do not show the same time points (10 h of 5 h respectively). This discrepancy needs to be avoided.
- 4) Fig. 2D-G: The authors deplete ALIX to inhibit repair. A control with CHMP recruitment need to be shown. According at least to the Hanson paper, a second ESCRT component needs to be co-depleted for an effect on repair. Validation of the depletion by Western blot is required.
- 5) Fig. 3AB: the recruitment data is somewhat trivial after Eapen et al., 2021. A negative control is missing. Why are different time points shown in A (p62) and B (other receptors)?
- 6) Fig. 3C-F: Again different time points are shown in C-D and E-F (5h versus 2h). The re-expression in Penta-KO cells was tested only at two hours, which is only when lysophagy begins (according to Fig. 2A). It is not clear, whether p62 and Nbr1 could do the job but only less efficiently. Re-expression levels need to be tested by western blotting. Labels should be "Flag-..." instead of "F-..." for clarity. In the text, the authors should use the terms "sufficient" and/or "necessary" to describe their finding with respect to autophagy receptors and lysophagy. Also, use "or" rather than "and" when single factors are described and not the combinations.
- 7) Fig. 4 shows the requirement and phosphorylation of TBK1. The involvement was largely demonstrated by Eapen and colleagues 2021 already. The difference is that this manuscript can now nail it down to lysophagy with the new approach and it is a nice validation of this approach. Why was shRNA used since the knockdown is inefficient according to Fig. S3? The BAPTA effect on TBK1 phosphorylation should be confirmed by Western blot.

Reviewer #2 (Comments to the Authors (Required)):

Shima and coworkers have submitted this manuscript entitled "The novel probe TMEM192-mKeima specifically assays lysophagy and reveals its initial steps" to be considered for publication as a Report in *JCB*. The authors have developed a novel probe, TMEM192-mKeima, to assay lysophagy (autophagic degradation of damaged lysosomes) which they show to more specifically report lysophagy than the existing Galectin-3 based probes. The latter probes are efficient in reporting lysosomal membrane rupture and is basically used in lysophagy studies to assay recovery from damage. The authors here show that the TMEM192mKeima assay effectively and specifically report lysophagic flux while the Gal-3 based probes also will report damage that is repaired by the ESCRT complex. Using the novel probe the authors also could show that TFEB is not required for lysophagy although important for lysosomal damage response since clearance of Gal-3 puncta is delayed by siRNA against TFEB. By knockout and reconstitution studies in pentaKO cells (HeLa cells with KO of p62, NBR1, NDP52, TAX1BP1 and OPTN) the author use the new probe to show that in this particular cell system only NDP52, TAX1BP1 and OPTN are able to rescue lysophagy when added back alone. They also show that TBK1 is required for lysophagy upon LLOMe treatment of cells.

This Report is very well written and presents a method which will be very important for researchers studying lysophagy. Most likely, this will be the method of choice for such studies. The authors show convincingly that this assay is more specifically and

directly reporting lysophagy and is useful to measure lysophagic flux relative to the existing Gal-3 based assays. As such, this paper is a "Tools" type of paper that describes a new method rather than a "Reports" type of paper, as I can see it from Journal of Cell Biology's instructions to authors. The roles of NDP52, TAX1BP1, OPTN and TBK1 in lysophagy has already been addressed very thoroughly by Eapen et al. 2021. Hence, the data presented here do not have the sufficient novelty for the standards of Reports papers in the JCB.

Some specific comments:

1. Except for IP experiments with FLAG-NDP52 to look for coIP with pTBK1 in HEK293 cells all the other experiments were carried out in HeLa cells. It would be more reassuring to see the probe behaving well in other cell lines too such as U2OS cells and MEFs and perhaps iNeurons which all are much better for microscopy analyses than the S3 variant of HeLa cells used here. The penta KO HeLa cells and its mother HeLa S3 cell line (HeLa CCL-2.2; ATCC) are not very well suited for microscopy.
2. The authors discuss that their conclusions differ somewhat from those of Eapen et al. 2021. However, in Eapen et al. more control experiments are performed and the heavy reliance on the pentaKO HeLa cells in the current study is not a strength. The pentaKO results should ideally be verified by using siRNA knockdown and reconstitution in other cell lines.
3. For the reader to be alerted to the fact that TMEM192 is not an arbitrarily chosen lysosomal transmembrane protein I think it would be pertinent to refer to the successful use in many labs and publications of the TMEM192-based LysoIP assay with reference to the 2017 Science paper by Abu-Remaileh et al. (PMID: 29074583).
4. The paper by Matsumoto et al. 2011 referenced to is not directly relevant as these authors then wrongly implicated Casein kinase 2 in the phosphorylation of S403 of p62/SQSTM1, not TBK1. The TBK1 phosphorylation of this site in p62/SQSTM1 was actually first reported by Pilli et al. 2012 (PMID: 22921120).

Reviewer #3 (Comments to the Authors (Required)):

In Shima et al the authors generate a new reporter of lysophagy that is based on expression of a TMEM192-mKeima fusion protein. The reporter localizes to lysosomes and exhibits a shift in fluorescence excitation (from 440 to 590nm) after induction of lysophagy, due to exposure of the pH-sensitive mKeima protein to the low pH environment of the lysosome lumen. The fluorescence shift can be visualized by light microscopy and the number of lysophagy-positive lysosomes per cell can be quantified. Using this reporter, the authors show that lysophagy occurs independently of TFEB and ESCRT proteins that are involved in lysosome biogenesis and repair, but requires numerous autophagy genes, STX17, and particular autophagy adaptors (OPTN, NDP52 and TAX1BP1). The authors further show that LLOMe-induced lysophagy requires the TANK-binding kinase 1 (TBK1), which is phosphorylated and recruits to lysosomes in a calcium-dependent manner in response to rupture.

This group previously developed the Gal3 assay for lysophagy, so the current study originates from the same group that previously established a critical assay in the field. As the Gal3 reporter does not directly measure lysosome turnover and is influenced by parallel pathways such as ESCRT-dependent membrane repair, the development of the new mKeima system is a needed advance. But while developing new methods to study lysophagy will have an impact on the field, the submitted work does not break significant new ground to further our understanding of the regulation of the process. Mechanistic studies mainly highlight the role of known or suspected players in validation of the new reporter, and the study also relies solely on non-physiologic lysosome damage induced by treatment with LLOMe. It seems that the authors could have leveraged the new reporter to identify more physiologic stressors that might trigger this process but such studies are not performed. It is also unclear that the reporter they have developed has the sensitivity that might be needed to support potential discovery-based studies. The overall lack of mechanistic insights developed in this study and the non-physiologic nature of the approaches limit the enthusiasm that the submitted work will have a significant impact on the field.

The following points could be addressed:

Major points:

1. It is unclear that the new Keima assay is working well or is nearly as sensitive as the Gal3 assay. For example in Figure 2A, more than 35 galectin spots per cell are induced in response to LLOMe, reporting significant lysosome rupture, but this leads to the appearance of only 7-8 mKeima spots per cell. While lysophagy may be only one of several parallel pathways that are induced in response to lysosome damage, it seems that more optimization or better quantification with the mKeima reporter is also needed. The authors describe a 590/440 fluorescence excitation ratio of greater than 1 as indicating a vesicle that is positive for lysophagy, but experiments demonstrating the rationale for this particular cutoff are not shown. Whether lysophagy induction leads to a shift that can be quantified from whole cell fluorescence, or whether the ratio that is chosen indicates the most sensitive setpoint from control experiments should be further considered and/or shown. As it stands, it would seem that lysophagy is a relatively minor player in response to LLOMe, and also that constitutive turnover in the absence of LLOMe is perhaps more significant, as indicated by mKeima fluorescence in untreated cells.
2. For the penta knockout cells, the authors should show if some aspect of lysosome function is impaired after attempted recovery from LLOMe-induced rupture. Also, the overexpression approach in Figure 3E and F to indicate which adaptors are crucial for lysophagy might be better investigated by pairing with loss of function studies, for example to examine if 3KO cells

(OPTN, NDP52 and TAX1BP1), if the authors have them, show equal impairment as 5KO.

3. The colocalization in Figure 4B between GFP-TBK1 and LAMP1 is not clear. Additional imaging with insets or perhaps imaging with enhanced resolution would help to clarify.

4. As the p-TBK1 puncta in Figure 4C are not numerous, some discussion of this seems warranted. Why are there only 1-1.5 of these particular lysosomes per cell?

5. Is the potential basal turnover indicated by the mKeima reporter in untreated cells also autophagy dependent?

6. The imaging in Figure 1D is not clear enough to discern what is concluded by the authors as a vesicle within a lysosome. Imaging with enhanced resolution would be needed to support this claim. As it is not central to the major points of the paper, this could also be removed.

Minor points:

1. In the last paragraph of the discussion the authors state that "Several pathways other than lysophagy have recently been reported to respond to lysosomal damage." This sentence should include more details or at least references.

2. Text edit- it seems Figure S3C should just be Figure S3.

Response to Reviewer Comments:

Thank you for the reviews of our manuscript # 202204048 entitled “The novel probe TMEM192-mKeima specifically assays lysophagy and reveals its initial steps “. We thank the reviewers for their constructive criticism and helpful comments that we have used to improve our paper. Below we have answered all the comments made by the reviewers.

Response to the comments of Editor

You will see that the reviewers express considerable interest in a lysophagy-specific reporter but also feel that the study in its current form does not provide a substantial advance in our understanding of the factors that are necessary or not essential for lysophagy. Other major concerns are whether TMEM192-mKeima is as sensitive as currently used Gal3-based assays and need for additional time points as well as assays in other cell types to better characterize the behavior and general utility of TMEM192-mKeima.

In both their reviews and private comments, the reviewers convey the sentiment that the study regrettably does not provide the kind of cutting-edge and highly novel findings that would be appropriate for the JCB Report format. However, given interest in the topic, we would be open to resubmission to JCB of a significantly revised and expanded manuscript for the Tools format. This revision would need to fully address the reviewers' concerns by providing a thorough characterization of TMEM192-mKeima as well as directly comparing it to relevant Gal3-based probes. As proof of concept of a lysophagy-specific reporter, a revised study should provide new insights into lysophagy that are not revealed by currently used Gal3-based assays. Please note that priority and novelty would be reassessed at resubmission. If you would like to resubmit this work to JCB, please contact the journal office to discuss an appeal of this decision or you may submit an appeal directly through our manuscript submission system.

Response:

We would like to resubmit our revised and expanded manuscript for the Tools format. In the revised manuscript, we have characterized the TMEM192-mKeima probe in more detail by examining multiple time points and using two cell lines as requested by the reviewers. By comparing the TMEM192-mKeima probe to the galectin-3 clearance assay, we showed that the TMEM192-mKeima probe is more specific to lysophagy than the galectin-3 assay. We also found some proteins which have been thought to be involved in lysophagy are important for the lysosomal damage response but not for lysophagy. In addition, we have performed a small-scale screen using this probe, and successfully identified novel factors that function in lysophagy. We hope that the current version fully addresses the reviewers' concerns and provides new insights

into lysophagy.

Point-by-Point response

Reviewer #1:

The molecular response to permeabilized or otherwise damaged lysosomes is very topical but still poorly understood. The pathway is of great interest because it is connected to various conditions including neurodegeneration in humans and represents a potential handle for cancer treatment. A major problem in the field is the differentiation of the two branches in quantitative assessments: lysosome repair on the one hand, or lysosome degradation through lysophagy on the other hand. A commonly used assay following Gal3, or a recent improvement with Gal3-mKeima (Eapen et al., 2021), is useful for many questions but cannot differentiate between the branches because both result in Gal3 being in an acidic environment. The authors now report on the development of cells that carry the mKeima reporter on the cytosolic tail of resident lysosomal protein TMEM192. Therefore, a fluorescence shift only occurs after autophagocytosis, but not upon repair and reacidification. The authors extensively characterize the cells providing evidence for the applicability of their approach and provide new findings and confirmations on the process of lysophagy.

The development of the cell line is elegant and very useful. The approach will be widely accepted and used in the field. Probably the most interesting finding using the approach in this manuscript is the proof of the differential timing of repair and lysophagy (Fig. 2A), which answers a long-standing question. The data on the involvement of autophagy receptors and on TBK1 is a great addition because they are conclusive albeit with somewhat limited scope because some of it was known already (most recently in Eapen et al., 2021, in Elife). Many extra time points, quantifications, statistics and controls are missing (see below) that need to be added before potential acceptance.

Major points:

1) Fig. 1B: The authors present only one time point. To characterize the cells and how the reporter behaves, they need to show images for more time points than just the 20h time point. They looked at the time course in Fig. 2A, but there the images are missing. Could a parallel stain with Gal3 be done, so to see the extent of damage. Are there any fluctuations in the signal? How does mKeima react to efflux of protons upon lysosome damage? It would also be nice to see red TMEM192-mKeima wrapped in LC3 for confirmation.

Response:

We thank the reviewer for the helpful comments. We have observed the cells expressing TMEM192-mKeima probe at 1h, 3h, 6h, 10h after LLOMe treatment (Fig.1 B and C) to characterize how the reporter behaves. The images for each time point are shown in Figure S2. Unfortunately, a parallel staining mKeima-TMEM192-mKeima with Gal3 or LC3 is technically difficult, because the unique characteristics of the fluorescent protein mKeima can only be applied in live cells (the pH gradient disappears in fixed cells). In addition, since the excitation and emission wavelengths of mKeima are similar to those of GFP and mCherry, it is difficult to obtain these signals simultaneously and observe fine co-localization of TMEM192-mKeima with Gal-3 or LC3. Instead, we observed the signals of TMEM192-mKeima and GFP-Gal3 separately in the cells expressing both probes (Fig. 2A and B).

How does mKeima react to efflux of protons upon lysosome damage?

Before LLOMe treatment, weak acidic TMEM192-mKeima signals were observed. This suggested that a little TMEM192-mKeima were delivered into lysosomes under basal conditions. Immediately after LLOMe, the signals disappeared because of loss of pH gradients in lysosomes. The brighter acidic TMEM192-mKeima signals came back in 1h after LLOMe treatment when the pH gradients in lysosomes were recovered. Since the background signal disappears once by LLOMe treatment, this makes easier to observe the mKeima signal delivered into lysosome (acidic mKeima) during lysosomal damage in this experimental condition.

2) *Fig. 2A: the corresponding images need to be shown. Statistic (errors and significance) should be shown. The time points before wash out would be important as reference. The entire time course should be repeated in at least one ATG-KO condition.*

Response:

We redid the experiment in control and FIP200 knockdown cells in Fig.2. We have repeated the experiment 3 times and shown all the images, statistic, and additional time points including before wash out (Fig. 2A and B).

3) *Fig. 2B-G: The authors compare effects on GFP-Gal3 and TMEM192-mKEIMA, but do not show the same time points (10 h of 5 h respectively). This discrepancy needs to be avoided.*

Response:

We apologize for the discrepancy. We redid the experiment and compared TMEM192-mKeima to GFP-galectin-3 at the same time point.

4) *Fig. 2D-G: The authors deplete ALIX to inhibit repair. A control with CHMP recruitment need to*

be shown. According at least to the Hanson paper, a second ESCRT component needs to be co-depleted for an effect on repair. Validation of the depletion by Western blot is required.

Response:

We attempted to perform TMEM192-mKeima assay in Alix and Tsg101 double knockdown cells, however, the cells were detached from the glass bottom dish by 1h-LLOMe treatment and it is difficult to observe the cells by live cell imaging. Although the previous studies showed that combined depletion of Alix and Tsg101 is required for an effect on repair, we observed clear delay in clearance of Gal-3 dots in Alix single knockdown cells in our condition. We validated the deletion of Alix by Western blot as shown below.

Fig.1 for Reviewers

HeLa cells expressing TMEM192-mKeima were treated with siRNA against Alix. The cells were bisected and subjected to microscopic analysis (Fig.2 E) or western blotting.

5) Fig. 3AB: the recruitment data is somewhat trivial after Eapen et al., 2021. A negative control is missing. Why are different time points shown in A (p62) and B (other receptors)?

Response:

In this manuscript, we moved the recruitment data to supplemental data section (Figure S5). Although Eapen et al previously reported the localization of OPTN, TAX1BP1 and CALCOCO2, we thought it is important to show the reasonable localization of the five adaptors for the subsequent rescue-experiment by re-expressing adaptors. We tested the localization of endogenous p62 and stably expressed GFP-adaptors under the same conditions with each control.

6) Fig. 3C-F: Again different time points are shown in C-D and E-F (5h versus 2h). The re-expression in Penta-KO cells was tested only at two hours, which is only when lysophagy begins (according to Fig. 2A). It is not clear, whether p62 and Nbr1 could do the job but only less efficiently. Re-expression levels need to be tested by western blotting.

Response:

We redid the experiments in Fig.3B-F and measured lysophagic activity at the same time point (10 h) when lysophagy was most active (Fig.1C). We have confirmed that the re-expression levels of the adaptors were approximately the same or higher than the endogenous by western blotting (Endogenous NBR1 was not detected with this NBR1 antibody, but FLAG-NBR1 was, so the amount of protein is

sufficient).

Fig.2 for Reviewers

WT or Penta-KO cells stably expressing TMEM192-mKeima and each FLAG-adaptor (p62, OPTN, NDP52, TAX1BP1, and NBR1) cultured in DMEM and then cell lysates were analyzed by immunoblotting.

To examine whether p62 is important for lysophagy, we performed the TMEM192-mKeima assay and the Gal-3 assay in p62 knockdown cells in addition to the re-expression experiment. We found that p62 was required for the lysosomal damage response, but not for lysophagy. We discuss this in the revised manuscript in page 11.

Labels should be "Flag-..." instead of "F-..." for clarity. In the text, the authors should use the terms "sufficient" and/or "necessary" to describe their finding with respect to autophagy receptors and lysophagy. Also, use "or" rather than "and" when single factors are described and not the combinations.

Response:

Thank you for correcting the words. We have rewritten them in this manuscript.

7) Fig. 4 shows the requirement and phosphorylation of TBK1. The involvement was largely demonstrated by Eapen and colleagues 2021 already. The difference is that this manuscript can now nail it down to lysophagy with the new approach and it is a nice validation of this approach. Why was shRNA used since the knockdown is inefficient according to Fig. S3? The BAPTA effect on TBK1 phosphorylation should be confirmed by Western blot.

Response:

In this manuscript, we investigated whether the targeting of TBK1 to lysosomes depends on Ca^{2+} to understand how TBK1 translocates to damaged lysosomes. We added the data showing that the targeting of TBK1 to lysosomes but not its activation was dependent on calcium in Fig.4 D, E and F.

This is firstly shown in this work. We used siRNA against TBK1 for better knockdown efficiency (Fig.4 A and B) and added a western blotting data to show the effect of BAPTA on TBK1 phosphorylation (Fig.4 F).

Reviewer #2

Shima and coworkers have submitted this manuscript entitled "The novel probe TMEM192-mKeima specifically assays lysophagy and reveals its initial steps" to be considered for publication as a Report in JCB. The authors have developed a novel probe, TMEM192-mKeima, to assay lysophagy (autophagic degradation of damaged lysosomes) which they show to more specifically report lysophagy than the existing Galectin-3 based probes. The latter probes are efficient in reporting lysosomal membrane rupture and is basically used in lysophagy studies to assay recovery from damage. The authors here show that the TMEM192mKeima assay effectively and specifically report lysophagic flux while the Gal-3 based probes also will report damage that is repaired by the ESCRT complex. Using the novel probe the authors also could show that TFEB is not required for lysophagy although important for lysosomal damage response since clearance of Gal-3 puncta is delayed by siRNA against TFEB. By knockout and reconstitution studies in pentaKO cells (HeLa cells with KO of p62, NBR1, NDP52, TAX1BP1 and OPTN) the author use the new probe to show that in this particular cell system only NDP52, TAX1BP1 and OPTN are able to rescue lysophagy when added back alone. They also show that TBK1 is required for lysophagy upon LLOMe treatment of cells.

This Report is very well written and presents a method which will be very important for researchers studying lysophagy. Most likely, this will be the method of choice for such studies. The authors show convincingly that this assay is more specifically and directly reporting lysophagy and is useful to measure lysophagic flux relative to the existing Gal-3 based assays. As such, this paper is a "Tools" type of paper that describes a new method rather than a "Reports" type of paper, as I can see it from Journal of Cell Biology's instructions to authors. The roles of NDP52, TAX1BP1, OPTN and TBK1 in lysophagy has already been addressed very thoroughly by Eapen et al. 2021. Hence, the data presented here do not have the sufficient novelty for the standards of Reports papers in the JCB.

Some specific comments:

1. Except for IP experiments with FLAG-NDP52 to look for coIP with pTBK1 in HEK293 cells all the other experiments were carried out in HeLa cells. It would be more reassuring to see the probe behaving well in other cell lines too such as U2OS cells and MEFs and perhaps iNeurons which all are much better for microscopy analyses than the S3 variant of HeLa cells used here. The penta KO

HeLa cells and its mother HeLa S3 cell line (HeLa CCL-2.2; ATCC) are not very well suited for microscopy.

Response:

We thank the reviewer for the helpful comments. We have added the data of TMEM192-mKeima assay in U2OS cells, showing that this novel assay is feasible in cells other than HeLa cells (Fig. 1 H and I). HeLa S3 cell line was used only for the reconstitution study in Fig. 3D and E, HeLa Kyoto cell line was used in the other experiments in this manuscript.

2. The authors discuss that their conclusions differ somewhat from those of Eapen et al. 2021. However, in Eapen et al. more control experiments are performed and the heavy reliance on the pentaKO HeLa cells in the current study is not a strength. The pentaKO results should ideally be verified by using siRNA knockdown and reconstitution in other cell lines.

Response:

As the reviewer pointed out, Eapen et al have already reported that TAX1BP1 are necessary and sufficient to promote lysophagy in HeLa cells. In this manuscript, we tested all five adaptors but focused a little more on p62, because conflicting findings have been obtained regarding the involvement of p62 in lysophagy. This was not discussed much in Eapen's paper 2021. To clarify this, we re-evaluated the involvement of all five autophagy receptors using TMEM192-mKeima probe, and showed that OPTN, NDFP52, and TAX1BP1, but not p62, are required for lysophagy (Fig. 3 D, E). As requested, we performed TMEM192-mKeima assay in knockdown cells using siRNA against each the adaptor in HeLa Kyoto cells (Fig.3F).

3. For the reader to be alerted to the fact that TMEM192 is not an arbitrarily chosen lysosomal transmembrane protein I think it would be pertinent to refer to the successful use in many labs and publications of the TMEM192-based LysoIP assay with reference to the 2017 Science paper by Abu-Remaileh et al. (PMID: 29074583).

Response:

We have described about TMEM192 in the text (page 5) with reference to the Science paper. " TMEM192 is a commonly used lysosomal marker, and has been utilized for imaging and immunoprecipitation of lysosomes (Abu-Remaileh et al., 2017) "

4. The paper by Matsumoto et al. 2011 referenced to is not directly relevant as these authors then wrongly implicated Casein kinase 2 in the phosphorylation of S403 of p62/SQSTM1, not TBK1. The

TBK1 phosphorylation of this site in p62/SQSTM1 was actually first reported by Pilli et al. 2012 (PMID: 22921120).

Response:

We apologize that we did not cite the appropriate paper. We cited the paper by Pilli et al. in the revised manuscript.

Reviewer #3

In Shima et al the authors generate a new reporter of lysophagy that is based on expression of a TMEM192-mKeima fusion protein. The reporter localizes to lysosomes and exhibits a shift in fluorescence excitation (from 440 to 590nm) after induction of lysophagy, due to exposure of the pH-sensitive mKeima protein to the low pH environment of the lysosome lumen. The fluorescence shift can be visualized by light microscopy and the number of lysophagy-positive lysosomes per cell can be quantified. Using this reporter, the authors show that lysophagy occurs independently of TFEB and ESCRT proteins that are involved in lysosome biogenesis and repair, but requires numerous autophagy genes, STX17, and particular autophagy adaptors (OPTN, NDP52 and TAX1BP1). The authors further show that LLOMe-induced lysophagy requires the TANK-binding kinase 1 (TBK1), which is phosphorylated and recruits to lysosomes in a calcium-dependent manner in response to rupture.

This group previously developed the Gal3 assay for lysophagy, so the current study originates from the same group that previously established a critical assay in the field. As the Gal3 reporter does not directly measure lysosome turnover and is influenced by parallel pathways such as ESCRT-dependent membrane repair, the development of the new mKeima system is a needed advance. But while developing new methods to study lysophagy will have an impact on the field, the submitted work does not break significant new ground to further our understanding of the regulation of the process. Mechanistic studies mainly highlight the role of known or suspected players in validation of the new reporter, and the study also relies solely on non-physiologic lysosome damage induced by treatment with LLOMe. It seems that the authors could have leveraged the new reporter to identify more physiologic stressors that might trigger this process but such studies are not performed. It is also unclear that the reporter they have developed has the sensitivity that might be needed to support potential discovery-based studies. The overall lack of mechanistic insights developed in this study and the non-physiologic nature of the approaches limit the enthusiasm that the submitted work will have a significant impact on the field.

The following points could be addressed:

Major points:

1. It is unclear that the new Keima assay is working well or is nearly as sensitive as the Gal3 assay. For example in Figure 2A, more than 35 galectin spots per cell are induced in response to LLOMe, reporting significant lysosome rupture, but this leads to the appearance of only 7-8 mKeima spots per cell. While lysophagy may be only one of several parallel pathways that are induced in response to lysosome damage, it seems that more optimization or better quantification with the mKeima reporter is also needed. The authors describe a 590/440 fluorescence excitation ratio of greater than 1 as indicating a vesicle that is positive for lysophagy, but experiments demonstrating the rationale for this particular cutoff are not shown. Whether lysophagy induction leads to a shift that can be quantified from whole cell fluorescence, or whether the ratio that is chosen indicates the most sensitive setpoint from control experiments should be further considered and/or shown. As it stands, it would seem that lysophagy is a relatively minor player in response to LLOMe, and also that constitutive turnover in the absence of LLOMe is perhaps more significant, as indicated by mKeima fluorescence in untreated cells.

Response:

We thank the reviewer for the critical comments. For analysis of acidic TMEM192-mKeima puncta, we used ImageJ software. We carefully set a threshold for each experiment because TMEM192-mKeima in lysosomes (594 nm) also showed weak signals at 445 nm excitation, and intact lysosomes that contain other damaged lysosomes also have their own TMEM192-mKeima signals at 445 nm excitation. The intensity at 594 nm was divided by the intensity at 445 nm, and if it was above the set threshold, puncta of mKeima were considered to be TMEM192-mKeima in an acidic environment.

2. For the penta knockout cells, the authors should show if some aspect of lysosome function is impaired after attempted recovery from LLOMe-induced rupture. Also, the overexpression approach in Figure 3E and F to indicate which adaptors are crucial for lysophagy might be better investigated by pairing with loss of function studies, for example to examine if 3KO cells (OPTN, NDP52 and TAX1BP1), if the authors have them, show equal impairment as 5KO.

Response:

As requested, we examined lysosome function during recovery from the damage using Magic red, which is widely used to evaluate lysosome function. As shown below, the intensity of Magic red was clearly lower in Penta KO cells compared to that of WT cells 5 h after washout of LLOMe, showing that lysosome function was impaired in penta KO cells during the recovery from LLOMe-induced damage.

Fig.3 for Reviewers

WT or Penta KO HeLa cells were cultured for 1 h in the presence of 1 mM LLOMe, followed by culture in DMEM medium containing Hoechst without LLOMe. After incubation for 5 h, cells were observed under a fluorescence microscope. Scale bars, 10 μ m.

As suggested, we performed loss of function studies using siRNAs against each adaptors or combination of three adaptors (OPTN, NDP52, and TAX1BP1) in cells expressing TMEM192-mKemia probe. Lysophagic activity was impaired in OPTN, NDP52, or TAX1BP1 knockdown cells, but not p62 and Nbr1 knockdown cells. A severer defect was observed in the triple knockdown cells (OPTN, NDP52, and TAX1BP1). These results are consistent with the experiments in which each adaptor was introduced into Penta KO cells in Fig.3D, E.

3. *The colocalization in Figure 4B between GFP-TBK1 and LAMP1 is not clear. Additional imaging with insets or perhaps imaging with enhanced resolution would help to clarify.*

Response:

We have replaced clearer images with insets in Fig. 4C in this manuscript.

4. As the p-TBK1 puncta in Figure 4C are not numerous, some discussion of this seems warranted. Why are there only 1-1.5 of these particular lysosomes per cell?

Response:

The p-TBK1 puncta were well observed in time dependent manner when FIP200 KO cells in which lysophagy does not progress beyond the initial step were used. This suggested that the localization of p-TBK1 on lysosomes was transient and difficult to observe in WT cells.

Fig. 4 for Reviewer

WT or FIP200 KO HeLa cells were treated with or without 1 mM LLOMe for 1 or 3 h, immunostained with p-TBK1 and LAMP1 antibodies, and then analyzed by fluorescence microscopy. Scale bars, 10 μ m.

5. Is the potential basal turnover indicated by the mKeima reporter in untreated cells also autophagy dependent?

Response:

The potential basal turnover appears to be both autophagy-dependent and autophagy-independent. A weak TMEM192-mKeima signals at 594 nm excitation was observed in untreated cells. Similar or slightly weaker signals were observed in FIP200 KO cells, suggesting that the signal at the basal level is autophagy-independent in part. The signals may represent invagination of lysosomal membrane through some other machinery such as MVB formation/microautophagy. We explained this in the text.

Fig. 5 for Reviewer.

WT and FIP200 KO HeLa cells stably expressing TMEM192-mKeima were cultured in DMEM containing Hoechst for 20 h and observed by fluorescence microscopy. Scale bars, 10 μ m.

6. *The imaging in Figure 1D is not clear enough to discern what is concluded by the authors as a vesicle within a lysosome. Imaging with enhanced resolution would be needed to support this claim. As it is not central to the major points of the paper, this could also be removed.*

Response:

We have included clearer fluorescence microscopy images with insets in Fig. 1D in the revised manuscript.

Minor points:

1. *In the last paragraph of the discussion the authors state that "Several pathways other than lysophagy have recently been reported to respond to lysosomal damage." This sentence should include more details or at least references.*

Response:

We have added more detailed descriptions and references in the revised manuscript page 19.

2. *Text edit- it seems Figure S3C should just be Figure S3.*

Response:

We thank the reviewer for the correction. It has been corrected.

June 4, 2023

Re: JCB manuscript #202204048R-A

Prof. Tamotsu Yoshimori
Osaka University
2-2 Yamadaoka, Suita
Osaka 565-0871
Japan

Dear Prof. Yoshimori,

Thank you for submitting your revised manuscript entitled "The novel probe TMEM192-mKeima specifically assays lysophagy and reveals its initial steps." The manuscript has been seen by the original reviewers whose full comments are appended below. While the reviewers continue to be overall positive about the potential utility of the TMEM192-mKeima reporter, some important issues remain.

As you will see Reviewers #1 and #3 express concerns about the sensitivity of TMEM192-mKeima. We feel that this tool could be useful in analyzing lysosome damage even if it needs to be used in combination with Gal3 in some circumstances. However, please address the reviewers' concerns regarding low sensitivity and dynamic ranges with new experimental data and/or thorough discussion to clearly indicate the limitations and caveats of TMEM192-mKeima. The normalization method and assay timing should be consistent between experiments (Rev#1, point 3). New experiments should also be performed to address the concern regarding the TRIM protein screen (Rev#1 point 6). As pointed out by the reviewer, the conclusions should either be strengthened by providing new data or negative statements should be toned down.

Our general policy is that papers are considered through only one revision cycle; however, given that the suggested changes are relatively minor we are open to one additional short round of revision.

Please submit the final revision in 1-3 months, along with a cover letter that includes a point by point response to the remaining reviewer comments.

Thank you for this interesting contribution to Journal of Cell Biology. You can contact me or the scientific editor listed below at the journal office with any questions, cellbio@rockefeller.edu or call (212) 327-8588.

Sincerely,

Xiaochen Wang, PhD
Monitoring Editor
Journal of Cell Biology

Dan Simon, PhD
Scientific Editor
Journal of Cell Biology

Reviewer #1 (Comments to the Authors (Required)):

This is a resubmission of a manuscript submitted a year ago that required major revision. At the time, the criticism was that the manuscript, while describing a useful new tool for analysis of lysophagy, did not provide enough novelty and advance in understanding of lysophagy. The authors have added more characterization such as timing and confirmation of 594 puncta inside of lysosomes. They also now provide new data in which they exploit the tool for the analysis of factors involved in lysophagy. However, these data are not conclusive and raise new concerns, also with regard to applicability of the new cell line as a tool. The new data as they stand therefore do not represent the expected major advancement in the field and leave questions open.

1) Referee #3 previously rightly raised the concern about the sensitivity of the assay using the TMEM192-mKeima cell line. This now becomes more apparent when the authors attempt to make negative statements for example regarding UBE2QC1. The authors see no effect on mKeima conversion upon UBE2QC1 depletion. This could well indicate an interesting lysophagy-independent function of UBE2QC1. However, knockdown was not verified and the expected reduction of K48 is actually not very strong, indicating that the depletion was not efficient. With low sensitivity, the assay may not pick up an effect on lysophagy,

particularly because UBE2QC1 was not tested along with the other E2s for comparison. At the current state, these results, although interesting, are not conclusive. This also applies to the other factors that did not score in the assay.

2) Along the same line, there is a discrepancy between the major effects on Gal3 clearance and low 594 puncta signals, or negative results for certain factors. This either implies a low sensitivity (drawing negative results into question) or an alternative (p62, UBE2LQ1, TFEB dependent) Gal3 clearance pathway that, however, remains unexplored in this work.

3) The sensitivity concern goes along with a normalization method that is difficult to trace and an assay timing that is very inconsistent between the experiments. This seems arbitrary. The authors also switch from 594 puncta/cell to "lysophagic activity" in the middle of the manuscript, which is not explained.

4) Another concern is that the authors tested the requirement of factors only at 1 mM LLOMe. Lysophagy requires healthy lysosomes. Depending on how many lysosomes are damaged, different factors may be required for lysophagy and different times are needed to acidify the TMEM192-mKeima sensor.

5) The identification of UBE2s involved in lysophagy is interesting, but it requires more work to confirm their function. The statement that UBE2L3 regulates K48 is misleading, because knockdown increases K48 which can only be explained through an indirect function.

6) The screen of TRIM proteins also seems preliminary. Only 1 siRNA was used per gene and the data are superficially evaluated. Statistics, z-score etc is missing.

Reviewer #2 (Comments to the Authors (Required)):

The authors have revised the paper in a very nice way and answered my comments/criticisms in a clearly satisfactory manner. I think this will be an important Tools paper receiving attention and citations from researchers studying lysophagy and lysosomal damage and repair pathways.

Reviewer #3 (Comments to the Authors (Required)):

The revised manuscript has been improved by the new experiments that were performed. This reviewer agrees that the research would be more suitable as a Tools style paper.

One concern that remains and wasn't addressed in the rebuttal is that the new reporter has a very poor dynamic range compared to GFP-Gal3, suggesting it is not able to report the majority of rupture/turnover events, and this may limit the utility of the assay moving forward. It seems that this should at least be discussed.

The data shown in Reviewer's Figure 3 with the Magic Red assay should be quantified from biological replicates and added to the paper.

Response to Reviewer Comments:

Thank you for the reviews of our manuscript # 202204048 entitled “The novel probe TMEM192-mKeima specifically assays lysophagy and reveals its initial steps “. We thank the reviewers for their helpful comments. Below we have answered all the comments made by the reviewers. The revisions are noted in colored letters in the manuscript.

Response to the comments of Editor

The manuscript has been seen by the original reviewers whose full comments are appended below. While the reviewers continue to be overall positive about the potential utility of the TMEM192-mKeima reporter, some important issues remain.

As you will see Reviewers #1 and #3 express concerns about the sensitivity of TMEM192-mKeima. We feel that this tool could be useful in analyzing lysosome damage even if it needs to be used in combination with Gal3 in some circumstances. However, please address the reviewers' concerns regarding low sensitivity and dynamic ranges with new experimental data and/or thorough discussion to clearly indicate the limitations and caveats of TMEM192-mKeima. The normalization method and assay timing should be consistent between experiments (Rev#1, point 3). New experiments should also be performed to address the concern regarding the TRIM protein screen (Rev#1 point 6). As pointed out by the reviewer, the conclusions should either be strengthened by providing new data or negative statements should be toned down.

Response:

Thank you so much for giving us the opportunity to revise the manuscript. In this revision, we have conducted an experiment to discuss the sensitivity of TMEM192-mKeima, which is a major concern of the reviewers. We added the new result in Fig S4C and discussed the sensitivity of TMEM192-mKeima in the response to Reviewer #1 (point 2) and Reviewer#3. However, as the reviewers pointed out, we could not exclude the possibility that the assay may not pick up an effect on lysophagy by knockdown of TFEB or p62 because of its low sensitivity and dynamic range. Therefore, we have described the limitations of TMEM192-mKeima in this manuscript. For ULB2QL1, we were inconclusive due to limitations of analysis method, and decided to exclude ULB2QL1 from the supplemental data. About the concern regarding the normalization method, assay timing, and the small-scaled screening, we explained in the response to Reviewer #2. We hope that this version addresses the concerns of the reviewers.

Point-by-Point response

Reviewer #1:

This is a resubmission of a manuscript submitted a year ago that required major revision. At the time, the criticism was that the manuscript, while describing a useful new tool for analysis of lysophagy, did not provide enough novelty and advance in understanding of lysophagy. The authors have added more characterization such as timing and confirmation of 594 puncta inside of lysosomes. They also now provide new data in which they exploit the tool for the analysis of factors involved in lysophagy. However, these data are not conclusive and raise new concerns, also with regard to applicability of the new cell line as a tool. The new data as they stand therefore do not represent the expected major advancement in the field and leave questions open.

1) Referee #3 previously rightly raised the concern about the sensitivity of the assay using the TMEM192-mKeima cell line. This now becomes more apparent when the authors attempt to make negative statements for example regarding UBE2QC1. The authors see no effect on mKeima conversion upon UBE2QC1 depletion. This could well indicate an interesting lysophagy-independent function of UBE2QC1. However, knockdown was not verified and the expected reduction of K48 is actually not very strong, indicating that the depletion was not efficient. With low sensitivity, the assay may not pick up an effect on lysophagy, particularly because UBE2QC1 was not tested along with the other E2s for comparison. At the current state, these results, although interesting, are not conclusive. This also applies to the other factors that did not score in the assay.

Response:

Thank you for the comment. To address this, we have attempted to confirm the knockdown efficiency of UBE2QL1 by qPCR since there are no commercial antibodies against UBE2QL1. We tested three individual siRNA against UBE2QL1 and three pairs of primers for qPCR. However, we could not detect a stable peak of UBE2QL1 even in control cells. Because the expression of other E2 enzymes (UBE2L3 and UBE2N) were detected and nicely knocked down, the expression level of UBE2QL1 seems to be low in HeLa Kyoto cells. Because we used the same sequence of siRNA against UBE2QL1 reported in Meyer's paper (Koerver et al, EMBO rep., 2019), and K48-ubiquitination was clearly suppressed in cells treated with the siRNA, we think the knockdown worked. We assume that the expression level of UBE2QL1 might be below the detection level in HeLa cells. Since we cannot provide data to conclude due to limitations of analysis methods, we decided to exclude supplemental Fig. S6 regarding UBE2QL1 from the manuscript. For the other factors that did not score in the

TMEM192-mKeima assay (Alix, TFEB, and p62), we confirmed that their expressions were nicely suppressed by siRNAs in protein levels (Fig.1 for Reviewers).

Fig.1 for Reviewers

HeLa cells were treated with siRNA against Alix, TFEB and p62. Cell lysates were analyzed by immunoblotting using the indicated antibodies.

2) Along the same line, there is a discrepancy between the major effects on Gal3 clearance and low 594 puncta signals, or negative results for certain factors. This either implies a low sensitivity (drawing negative results into question) or an alternative (p62, UBE2LQ1, TFEB dependent) Gal3 clearance pathway that, however, remains unexplored in this work.

Response:

In this revision, we have conducted an experiment to discuss the sensitivity of TMEM192-mKeima, which is a major concern of the reviewers. We directly demonstrated that the GFP-Gal3 assay monitors multiple lysosomal damage response pathways by comparing the effect of knockdown of FIP200 (lysophagy), Alix (ESCRT pathway), TFEB (TFEB pathway) or all three of them (we added the data in Figure S4C and described in page 9 line 23). The results clearly showed that lysophagy is only one of several parallel pathways monitored by Gal-3 assay, which could explain why the signal of TMEM192-mKeima is low compared to Gal-3 dots. The results indicate that the recovery of damaged lysosomes seems to be contributed by lysophagy, Alix, and TFEB pathways to the same degree, and other pathways also may be involved. Therefore, the small number of acidic TMEM192-mKeima may not be so unreasonable (about 50 Gal-3 dots per cell were induced by LLOMe in WT cells, while about 5 ~10 mKeima dots per cell were observed 10 h after wash out in Fig 2).

In this revision, we showed that knockdown of ALIX or TFEB suppressed the clearance of Gal-3 dots to the same degree as FIP200 (Fig.S4C). Because the TMEM192-mKeima probe detected the suppression of lysophagy in FIP200 knockdown cells and the experiments were performed under the same conditions, it is unlikely that TMEM192-mKeima failed to evaluate lysophagic activity in ALIX or TFEB knockdown cells due to its low sensitivity. We also believe that it is important to warn that it is risky to evaluate lysophagy by the Gal-3 assay alone.

Even if the sensitivity of TMEM192-mKeima is lower than Gal-3, we demonstrated that the sensitivity of TMEM192-mKeima is enough to evaluate the factors involved in lysophagy such as the adaptors and TBK1 as shown in Fig. 3 and Fig. 4, and is able to use for a screen in Fig. 6. These results suggest that TMEM192-mKeima should be a useful tool to study the molecular mechanism of lysophagy.

However, as the reviewers pointed out, we could not exclude the possibility that the assay may not pick up an effect on lysophagy by knockdown of TFEB or p62 because of its low sensitivity and dynamic range. Therefore, we have described the limitations of TMEM192-mKeima in this manuscript (revisions are noted in red. page17 line 4, page 19, line 6).

We added the data in Fig. S4C, which shows that the GFP-Gal3 assay monitors multiple lysosomal damage response pathways (lysophagy, Alix pathway, TFEB pathway, and others).

HeLa cells stably expressing GFP-Gal3 were treated with siRNA against FIP200, Alix, TFEB or treated with control siRNA. After siRNA treatment, cells were cultured for 1 h in the presence of 1 mM LLOMe, followed by culture in DMEM without LLOMe. After 10 h, cells were observed by fluorescence microscopy. Scale bars, 10 μ m. The graph shows the percentage of Gal-3 puncta

remaining after 10 h of incubation. More than 30 cells were analyzed per condition in each experiment. Error bars indicate means \pm SEM (n = 3). ****P < 0.01** (one-way ANOVA followed by Dunnett's test).

We have also attempted to show that TFEB does not play a major role in lysophagy by another experiment besides the TMEM192-mKeima assay. ULK1 is one of autophagy-related proteins which is required for autophagosome-formation and often used as a marker for autophagosome-formation site. When lysophagy is induced, ULK1 is observed as puncta on lysosomes (Maejima et al., EMBO J., 2013). We examined whether autophagosome formation was affected by the deletion of TFEB. The results showed that the formation of ULK1 puncta was not affected (Fig. R2), suggesting that TFEB is not involved in at least the early step of autophagosome formation.

Fig. R2 for Reviewers

ULK1/2 DKO HeLa cells stably expressing mNeonGreen -ULK1 were treated with siRNA against FIP200, TFEB, or control siRNA. After 48 h of siRNA treatment, cells were cultured with or without 1 mM LLOMe for 1 h, immunostained with LAMP1 antibody, and then analyzed by fluorescence microscopy. Scale bars, 10 μ m. The graph shows the percentage of ULK1-positive LAMP1 dots. More than 30 cells were analyzed per condition in each experiment. Error bars indicate means \pm SEM (n = 3). ***P < 0.001, NS: not significant (one-way ANOVA followed by Dunnett's test).

3) The sensitivity concern goes along with a normalization method that is difficult to trace and an assay timing that is very inconsistent between the experiments. This seems arbitrary. The authors also switch from 594 puncta/cell to "lysophagic activity" in the middle of the manuscript, which is not explained.

Response:

In the main figures, we have conducted the TMEM192-mKeima assay under the same conditions including assay timing; HeLa cells were cultured for 1 h in the presence of 1 mM LLOMe. After LLOMe was removed, cells were cultured in DMEM for 10 h and observed by fluorescence microscopy in live. However, with some siRNA, the cells were detached from the glass-bottom dishes by the medium-change process. Only in these cases, we changed the conditions; cells were cultured for 6 h in the presence of 1 mM LLOMe and observed by fluorescence microscopy in live. To ensure that this condition is appropriate, we confirmed that lysophagic activity is similar between the cells cultured for 5 h without LLOMe after LLOMe treatment for 1 h and the cells cultured for 6 h with LLOMe. All experiments contained FIP200-KD/KO cells as a control.

In this manuscript, we have unified the quantitative results to be presented in acidic puncta/cell. Only in Fig.6 A, we used total intensity of mKeima signal at 594 nm for the calculation because of the large number of the factors.

In response to the reviewer's concerns regarding the normalization, we have described below the details of the quantification method for the TMEM192-mKeima assay.

Fiji (<https://imagej.net/Fiji>) was used in the image analysis of the TMEM192-mKeima assay.

↓ The number of cells is counted by staining the nuclear with Hoechst® 33342.

↓ The mKeima signals at the excitation wavelength of 440 nm (neutral mKeima) are extracted by "image>Adjust>Threshold".

↓ The location of each the neutral mKeima signals is marked by "Analyze>Analyze particle".

↓ With the location information, the fluorescence intensity of each mKeima signals at 440 nm and 586

nm is measured by "ROI manager".

↓ The value of the ratio of the 586 nm mKeima signal intensity/440 nm mKeima signal intensity at each position are calculated.

↓ The value with a clear difference compared to the signal intensity ratio in LLOMe-untreated (basal) and FIP200-deficient conditions (lysophagy does not occur) is determined and defined as "acidic mKeima puncta (\approx lysosomes in which lysophagy has occurred)". Lysophagy activity is evaluated by calculating the number of "acidic mKeima puncta" per cell.

4) Another concern is that the authors tested the requirement of factors only at 1 mM LLOMe. Lysophagy requires healthy lysosomes. Depending on how many lysosomes are damaged, different factors may be required for lysophagy and different times are needed to acidify the TMEM192-mKeima sensor.

Response:

We performed the Gal-3 clearance assay and the TMEM192-mKeima assay under the same conditions (1 h treatment of 1 mM LLOMe) in Fig. 2 and Fig. 3, and found that Alix, TFEB, and p62 scored positive in the Gal-3 clearance assay but negative in the TMEM192-mKeima assay. Because lysophagy was suppressed by knocking-down of these factors in the Gal-3 assay, we assume that we conducted the TMEM192-mKeima assay under conditions where the factors were required. Our observation is consistent with the previous reports showing that the repair of damaged lysosomes by the ESCRT machinery is independent of lysophagy (Skowyra et al., Science, 2018, Radulovic et al., EMBO J, 2018). Harper's group also reported that p62 is not involved in lysophagy using their Lyso-mKeima assay (Eapen et al, Elife, 2021). Since studying lysophagy with 1 mM LLOMe in HeLa cells is a rather common protocol, (Eapen et al, Elife, 2021, Eapen et al., DOI: dx.doi.org/10.17504/protocols.io.bx8qprvw, Skowyra et al., Science, 2018), we think that the experimental conditions are appropriate.

5) The identification of UBE2s involved in lysophagy is interesting, but it requires more work to confirm their function. The statement that UBE2L3 regulates K48 is misleading, because knockdown increases K48 which can only be explained through an indirect function.

Response:

We identified UBE2N and UBE2L3 as E2 enzymes required for lysophagy using the TMEM192-mKeima probe. UBE2N and UBE2L3 were translocated to lysosomes upon LLOMe treatment, and their suppression affected ubiquitination of lysosomes, which strongly suggest their involvement in lysophagy. Since this is a tool paper, we only showed the involvement of those factors and detailed

functional analyses will be a future work.

We apologize the statement that UBE2L3 regulates K48 is misleading. The regulation of K48-Ub of lysosomes is complicated, because not only K48-ubiquitination but also K48-deubiquitination is thought to be required for lysophagy (Papadopoulos et al, EMBO J, 2017). It has been reported that p97/YOD1/UBXD1/PLAA complex removes K48-Ub from damaged lysosomes to promote autophagosome formation. Knockdown of the component of the complex resulted in increase of K48-Ub. Moreover, translocation of the complex to lysosome is dependent on ubiquitination of lysosomes (Koerver et al, EMBO rep, 2019). Assuming that multiple E2 enzymes involved in the ubiquitination of damaged lysosomes, it is difficult to predict how UBE2L3 knockdown lead to accumulation of K48-Ub. It could be direct or indirect. We have rewritten the statement that UBE2L3 regulates K48 as follows;

“we identified UBE2L3 and UBE2N as being required for lysophagy and somehow involved in K48-linked and K63-linked ubiquitination of lysosomes, respectively.” (page 14, line 20)

“UBE2L3 and UBE2N were shown to be involved in lysophagy and ubiquitination of lysosomes, although their function in lysophagy is not clear yet.” (page 16, line 14)

6) The screen of TRIM proteins also seems preliminary. Only 1 siRNA was used per gene and the data are superficially evaluated. Statistics, z-score etc is missing.

Response:

We performed a small-scale screen using siRNAs against TRIMs to show that TMEM192-mKeima a tool which can be applied to screen. In Fig.6A, primary screen was performed with one siRNA each because of the large number of the factors. In Fig.6B and Fig6C, different siRNAs against TRIM10, 16 or 27 were used to validate the results of Fig.6A. Thus, we have examined that lysophagy was defective by knockdown with two different siRNAs for TRIM10, 16, 27. Robust z-scores were calculated for LLOMe-treated samples for each deletion.

Reviewer #2

The authors have revised the paper in a very nice way and answered my comments/criticisms in a clearly satisfactory manner.

I think this will be an important Tools paper receiving attention and citations from researchers studying lysophagy and lysosomal damage and repair pathways.

Response:

We appreciate the reviewer's positive comments.

Reviewer #3

The revised manuscript has been improved by the new experiments that were performed. This reviewer agrees that the research would be more suitable as a Tools style paper.

One concern that remains and wasn't addressed in the rebuttal is that the new reporter has a very poor dynamic range compared to GFP-Gal3, suggesting it is not able to report the majority of rupture/turnover events, and this may limit the utility of the assay moving forward. It seems that this should at least be discussed.

Response:

This will be the same as the response to Editor and Reviewer #1 (point 2).

In this revision, we have conducted an experiment to discuss the sensitivity of TMEM192-mKeima, which is a major concern of the reviewers. We directly demonstrated that the GFP-Gal3 assay monitors multiple lysosomal damage response pathways by comparing the effect of knockdown of FIP200 (lysophagy), Alix (ESCRT pathway), TFEB (TFEB pathway) or all three of them (Figure S4C). The results clearly showed that lysophagy is only one of several parallel pathways monitored by Gal-3 assay, which could explain why the signal of TMEM192-mKeima is low compared to Gal-3 dots. The results indicate that the recovery of damaged lysosomes seems to be contributed by lysophagy, Alix, and TFEB pathways to the same degree, and other pathways also may be involved. Therefore, the small number of acidic TMEM192-mKeima may not be so unreasonable (about 50 Gal-3 dots per cell were induced by LLOMe in WT cells, while about 5 ~10 mKeima dots per cell were observed 10 h after wash out in Fig2).

In this revision, we showed that knockdown of ALIX or TFEB suppressed the clearance of Gal-3 dots to the same degree as FIP200 (Fig.S4C). Because the TMEM192-mKeima probe detected the suppression of lysophagy in FIP200 knockdown cells and the experiments were performed under the same conditions, it is unlikely that TMEM192-mKeima failed to evaluate lysophagic activity in ALIX or TFEB knockdown cells due to its low sensitivity. We also believe that it is important to warn that it is risky to evaluate lysophagy by the Gal-3 assay alone.

Even if the sensitivity of TMEM192-mKeima is lower than Gal-3, we demonstrated that the sensitivity

of TMEM192-mKeima is enough to evaluate the factors involved in lysophagy such as the adaptors and TBK1 as shown in Fig. 3 and Fig. 4, and is able to use for a screen in Fig. 6. These results suggest that TMEM192-mKeima should be a useful tool to study the molecular mechanism of lysophagy.

However, as the reviewers pointed out, we could not exclude the possibility that the assay may not pick up an effect on lysophagy by knockdown of TFEB or p62 because of its low sensitivity and dynamic range. Therefore, we have described the limitations of TMEM192-mKeima in this manuscript (revisions are noted in red. page17 line 4, page 19, line 6).

The data shown in Reviewer's Figure 3 with the Magic Red assay should be quantified from biological replicates and added to the paper.

Response:

The results of the Magic Red recovery assay were quantified from biological replicates (n=3) and added to Fig.S6.

August 31, 2023

RE: JCB Manuscript #202204048RR

Prof. Tamotsu Yoshimori
Osaka University
2-2 Yamadaoka, Suita
Osaka 565-0871
Japan

Dear Prof. Yoshimori,

Thank you for submitting your revised manuscript entitled "The novel probe TMEM192-mKeima specifically assays lysophagy and reveals its initial steps." We would be happy to publish your paper in JCB pending final revisions necessary to meet our formatting guidelines (see details below).

A. MANUSCRIPT ORGANIZATION AND FORMATTING:

1) Text limits: Character count for Tools is < 40,000, not including spaces. Count includes title page, abstract, introduction, results, discussion, and acknowledgments. Count does not include materials and methods, figure legends, references, tables, or supplemental legends.

2) Figure formatting: Tools may have up to 10 main text figures. Scale bars must be present on all microscopy images, including inset magnifications. Molecular weight or nucleic acid size markers must be included on all gel electrophoresis. Please add scale bars for the magnifications in Figures 1D, 4C, & 5B.

Also, please avoid pairing red and green for images and graphs to ensure legibility for color-blind readers. If red and green are paired for images, please ensure that the particular red and green hues used in micrographs are distinctive with any of the colorblind types. If not, please modify colors accordingly or provide separate images of the individual channels.

3) Statistical analysis: Error bars on graphic representations of numerical data must be clearly described in the figure legend. The number of independent data points (n) represented in a graph must be indicated in the legend. Please, indicate whether 'n' refers to technical or biological replicates (i.e. number of analyzed cells, samples or animals, number of independent experiments). If independent experiments with multiple biological replicates have been performed, we recommend using distribution-reproducibility SuperPlots (please see Lord et al., JCB 2020) to better display the distribution of the entire dataset, and report statistics (such as means, error bars, and P values) that address the reproducibility of the findings.

Statistical methods should be explained in full in the materials and methods. For figures presenting pooled data the statistical measure should be defined in the figure legends. Please also be sure to indicate the statistical tests used in each of your experiments (both in the figure legend itself and in a separate methods section) as well as the parameters of the test (for example, if you ran a t-test, please indicate if it was one- or two-sided, etc.). Also, if you used parametric tests, please indicate if the data distribution was tested for normality (and if so, how). If not, you must state something to the effect that "Data distribution was assumed to be normal but this was not formally tested."

4) Title: for conciseness we suggest the following slightly modified title: "The TMEM192-mKeima probe specifically assays lysophagy and reveals its initial steps."

5) Materials and methods: Should be comprehensive and not simply reference a previous publication for details on how an experiment was performed. Please provide full descriptions (at least in brief) in the text for readers who may not have access to referenced manuscripts. The text should not refer to methods "...as previously described."

6) For all cell lines, vectors, constructs/cDNAs, etc. - all genetic material: please include database / vendor ID (e.g., Addgene, ATCC, etc.) or if unavailable, please briefly describe their basic genetic features, even if described in other published work or gifted to you by other investigators (and provide references where appropriate). Please be sure to provide the sequences for all of your oligos: primers, si/shRNA, RNAi, gRNAs, etc. in the materials and methods. You must also indicate in the methods the source, species, and catalog numbers/vendor identifiers (where appropriate) for all of your antibodies, including secondary. If

antibodies are not commercial, please add a reference citation if possible.

7) Microscope image acquisition: The following information must be provided about the acquisition and processing of images:

- a. Make and model of microscope
- b. Type, magnification, and numerical aperture of the objective lenses
- c. Temperature
- d. Imaging medium
- e. Fluorochromes
- f. Camera make and model
- g. Acquisition software
- h. Any software used for image processing subsequent to data acquisition. Please include details and types of operations involved (e.g., type of deconvolution, 3D reconstitutions, surface or volume rendering, gamma adjustments, etc.).

8) References: There is no limit to the number of references cited in a manuscript. References should be cited parenthetically in the text by author and year of publication. Abbreviate the names of journals according to PubMed.

9) Supplemental materials: There are strict limits on the allowable amount of supplemental data. Tools may have up to 5 supplemental figures and 10 videos. You currently exceed this limit but, in this case, we will be able to give you the extra space. Please also note that tables, like figures, should be provided as individual, editable files. A summary of all supplemental material should appear at the end of the Materials and methods section. Please include one brief sentence per item.

10) eTOC summary: A ~40-50 word summary that describes the context and significance of the findings for a general readership should be included on the title page. The statement should be written in the present tense and refer to the work in the third person. It should begin with "First author name(s) et al..." to match our preferred style.

11) Conflict of interest statement: JCB requires inclusion of a statement in the acknowledgements regarding competing financial interests. If no competing financial interests exist, please include the following statement: "The authors declare no competing financial interests." If competing interests are declared, please follow your statement of these competing interests with the following statement: "The authors declare no further competing financial interests."

12) A separate author contribution section is required following the Acknowledgments in all research manuscripts. All authors should be mentioned and designated by their first and middle initials and full surnames. We encourage use of the CRediT nomenclature (<https://casrai.org/credit/>).

13) ORCID IDs: ORCID IDs are unique identifiers allowing researchers to create a record of their various scholarly contributions in a single place. Please note that ORCID IDs are required for all authors. At resubmission of your final files, please be sure to provide your ORCID ID and those of all co-authors.

14) JCB requires authors to submit Source Data used to generate figures containing gels and Western blots with all revised manuscripts. This Source Data consists of fully uncropped and unprocessed images for each gel/blot displayed in the main and supplemental figures. Since your paper includes cropped gel and/or blot images, please be sure to provide one Source Data file for each figure that contains gels and/or blots along with your revised manuscript files. File names for Source Data figures should be alphanumeric without any spaces or special characters (i.e., SourceDataF#, where F# refers to the associated main figure number or SourceDataFS# for those associated with Supplementary figures). The lanes of the gels/blots should be labeled as they are in the associated figure, the place where cropping was applied should be marked (with a box), and molecular weight/size standards should be labeled wherever possible. Source Data files will be directly linked to specific figures in the published article.

15) Journal of Cell Biology now requires a data availability statement for all research article submissions. These statements will be published in the article directly above the Acknowledgments. The statement should address all data underlying the research presented in the manuscript. Please visit the JCB instructions for authors for guidelines and examples of statements at (<https://rupress.org/jcb/pages/editorial-policies#data-availability-statement>).

B. FINAL FILES:

-- High-resolution figure and MP4 video files: See our detailed guidelines for preparing your production-ready images,

<https://jcb.rupress.org/fig-vid-guidelines>.

Thank you for this interesting contribution, we look forward to publishing your paper in Journal of Cell Biology.

Sincerely,

Xiaochen Wang, PhD
Monitoring Editor
Journal of Cell Biology

Dan Simon, PhD
Scientific Editor
Journal of Cell Biology